# Multi-LLM Debate: Framework, Principals, and Interventions

**Andrew Estornell**
ByteDance Research
andrew.estornell@bytedance.com

**Yang Liu**
University of California, Santa Cruz
yangliu@ucsc.edu

## Abstract

The flexible and generalized nature of large language models has allowed for their application in a wide array of language-based domains. Much like their human contemporaries, these models are capable of engaging in discussions and debates as a means of improving answer quality. We first take a theoretical approach to analyzing debate and provide a framework through which debate can be mathematically examined. Building on this framework, we provide several theoretical results for multi-agent debate. In particular, we demonstrate that similar model capabilities, or similar model responses, can result in static debate dynamics where the debate procedure simply converges to the majority opinion. When this majority opinion is the result of a common misconception (possibly ingrained in the models through shared training data) debate is likely to converge to answers associated with that common misconception. Using insights from our theoretical results, we then propose three interventions that improve the efficacy of debate. For each intervention, we provide theoretical results demonstrating how debate is improved. We also demonstrate that these interventions result in better performance on four common benchmark tasks.

## 1   Introduction

Large language models (LLMs) have demonstrated a remarkable ability to perform unseen tasks with high efficacy. This behavior, often referred to as emergent, allows LLMs to serve as general-purpose tools for a wide array of language-based functions. One such behavior of particular interest is the ability of LLMs to intake and process opinions from other models (or humans). As shown in several previous works, this ability allows LLMs to collaboratively solve tasks by engaging in *debate* Chan et al. [2023], Liang et al. [2023], Du et al. [2023]. For a given task, multi-agent debate operates by eliciting responses from each model, distributing those responses among the models, and then eliciting updated responses from each model.

In this work, we aim to explore the debate procedure, by first providing a theoretical framework through which debate can be better understood. This framework draws inspiration from Bayesian inference and in-context learning, showing that debate can be partially viewed as a special type of in-context learning. Through this framework, we then provide several theoretical insights into the debate procedure. In particular, we demonstrate the susceptibility of multi-agent debate to echo-chamber effects. These echo-chamber effects are especially consequential when they stem from a shared misconception between a majority of models, which can arise from circumstances such as highly correlated training data between each model.

We then leverage results from our theoretical framework to improve the efficacy of the debate procedure. In particular, we propose three *interventions* (modifications to the debate procedure). First, *diversity-pruning* which aims to maximize the information entropy in model responses at each round of debate; this intervention has the added benefit of reducing the severity of the echo chamber

effect. Second, *quality-pruning* which aims to maximize the relevance of each model's response. We demonstrate that this intervention improves the likelihood that the debate procedure converges to correct answers. Lastly, *misconception-refutation* which directly identifies, and attempts to refute misconceptions in model responses. This intervention takes inspiration from works such as Robinson et al. [2022] which demonstrate that LLMs are often more skilled at *evaluating* answers, compared with directly providing answers. For each of our interventions, we provide theoretical results outlining the way in which each improves debate. We also conduct experiments on four common benchmarks demonstrating that these interventions improve debate efficacy in practice.

**Our contributions** 1) We propose a theoretical framework for multi-LLM debate that draws on connections from in-context learning and Bayesian inference. 2) We provide theoretical insights on several key principles of multi-LLM debate. 3) Using these insights, we design three debate interventions which result in consistent improvement to debate, across four language benchmarks (BoolQ, MMLU, MathQ, TruthfuQA) and three families of models (GPT, Lama, and Mistral).

## 2 Related Work

Our work is closely related to multi-agent debate, which focuses on iterative collaboration between agents to make a decision Chan et al. [2023], Liang et al. [2023], Du et al. [2023], Khan et al. [2024], Irving et al. [2018], Michael et al. [2023], Rasal [2024], Pham et al. [2023], Chang [2024b]. These works often focus on multi-agent debate in the context of question-answering tasks and aim to provide higher quality answers (compared to those of a single model) by engaging multiple models in discussion. The preliminary debate framework, proposed by Du et al. [2023], facilitates debate by first asking each model a question, and then iteratively re-asking agents that same question contextualized by the responses of all models in the previous round. Different variants of this procedure have been proposed: debate where models have different functionality Liang et al. [2023], round-robin style debate Chan et al. [2023], dynamically controlling the level of disagreement between agents in debate Chang [2024a], or using judges to assess the correctness of debaters Khan et al. [2024]. Other techniques for iteratively improving the quality of answers have also been proposed, e.g., chain-of-thought Wei et al. [2022], Kojima et al. [2022], self-consistency Wang et al. [2022], Singhal et al. [2023], and self-reflection Ren et al. [2023].

Similar to debate, there have been investigations into the use of different LLMs to engage with one-another Liu et al. [2023], Abdelnabi et al. [2023], Zhang et al. [2023], Li et al. [2023c], Park et al. [2023a], explain their rational to others Wang et al. [2023a], or collaboratively engage in general tasks Li et al. [2023a], Wang et al. [2023b], Park et al. [2023b], Wu et al. [2023], Hong et al. [2023], Li et al. [2023d,b], Tsao and AILAB [2023]. While debate has shown promise in a wide range of domains, several works have also demonstrated that the debate process can be unstable and can lead to worse performance than using just a single model Wang et al. [2024], Smit et al. [2023].

Our work is also related to in-context learning and Bayesian inference. The former, Brown et al. [2020], Min et al. [2022], Lampinen et al. [2022] demonstrates that LLMs can perform unseen tasks when provided only a few examples of that task. Other works Xie et al. [2021], Jiang [2023] have shown a connection between in-context learning and Bayesian inference; the additional examples provided to the model can be viewed as updates to the model's posterior distribution over tokens.

## 3 Preliminaries

**Debate**   Let $\mathbf{x}$ be a given question, with associated answer $\mathbf{y}$, for example $\mathbf{x} =$ *"What color is the sky?"* and $\mathbf{y} =$ *"Blue"*. Following the debate procedure proposed by Du et al. [2023], a collection of $n$ LLMs (also referred to as agents) collaborate to infer the correct answer $\mathbf{y}$ by iteratively engaging in discussion over $T$ rounds, as described next:

- At round $t = 0$ each agent $i$ observes task $\mathbf{x}$, then provides response $\mathbf{z}_i^{(0)}$.

- At rounds $t > 0$ each agent $i$ observes task $\mathbf{x}$ and the outputs of the $n$ agents at the previous timesetep $Z^{(t-1)} = (\mathbf{z}_1^{(t-1)}, \ldots, \mathbf{z}_n^{(t-1)})$, then provides response $\mathbf{z}_i^{(t)}$.

- The debate process ends if $t = T$ or if agents reach a consensus.

To measure if consensus is reached a function $a$ extracts an *answer*[1] from a given response $\mathbf{z}$. Suppose $\mathbf{z} =$ *"During the day, the sky is blue"*, then $a(\mathbf{z}) =$ *"Blue"*. At round $t$, the probability that agent $i$ provides response $\mathbf{z}_i^{(t+1)}$, is given by

$$\mathbb{P}_{\text{model}}\left( \underbrace{\mathbf{z}_i^{(t+1)}}_{\text{updated model response}} \Big| \underbrace{\mathbf{x}}_{\text{task}}, \underbrace{Z^{(t)} = (\mathbf{z}_1^{(t)}, \dots, \mathbf{z}_n^{(t)})}_{\text{all model responses from previous round}}, \underbrace{\phi_i}_{\text{model parameters}} \right) \qquad (1)$$

where $\phi_i$ captures model hyperparameters (such as training data, architecture, etc.). Both the input $(Z^{(t)}, \mathbf{x})$ as well as the hyperparameters $\phi_i$, ultimately influence the output $\mathbf{z}_i^{(t+1)}$. Note that on each round, all agents observe the *same* input, namely $(Z^{(t)}, \mathbf{x})$. Thus, differences in agent output $\mathbf{z}_i^{(t+1)}$ are determined by both the stochastic nature of output generations, and the unique parameters $\phi_i$ of each model. For notational convenience, we drop the subscript in $\mathbb{P}_{\text{model}}$ when the parameters $\phi_i$ are given, and simply write $\mathbb{P}(\cdot | \phi_i)$.

The key distinction between our approach and "vanilla" debate, is that we will leverage latent concepts (discussed next) to modify the responses in $Z^{(t)}$ in between each round of debate.

**Latent Concepts**  Central to our investigation is the idea of *latent concepts* in language generation. As outlined by Xie et al. [2021], Jiang [2023] latent concepts capture the idea that language is not generated at random (either by humans or by models). Rather, when generating language, we first have an idea or an intention form in our minds; we then select words that we believe will convey that underlying idea or intention. More formally, let $\Theta$ be a *latent concept space* and let $\boldsymbol{\theta} \in \Theta$ be a *concept*. Following Xie et al. [2021], tasks $\mathbf{x}$, and their associated answer $\mathbf{y}$ are generated by first selecting a vector of latent concept $\boldsymbol{\theta} \in \Theta$ and then sampling $(\mathbf{x}, \mathbf{y}) \sim D(\boldsymbol{\theta})$, where $D$ is some distribution mapping concepts to task-answer pairs. Similarly, when providing responses, models will observe $\mathbf{x}$, and infer the latent concept $\boldsymbol{\theta}$, or more generally a distribution over the latent concept space, and then generate a response according to those inferred concepts, i.e., the model's generation probability in Equation 1 can be expressed as

$$\mathbb{P}\big(\mathbf{z}_i^{(t+1)} \big| \mathbf{x}, Z^{(t)}, \phi_i\big) = \prod_{\boldsymbol{\theta} \in \Theta} \mathbb{P}\big(\mathbf{z}_i^{(t+1)} \big| \boldsymbol{\theta}, \mathbf{x}, Z^{(t)}, \phi_i\big) \mathbb{P}\big(\boldsymbol{\theta} \big| \mathbf{x}, Z^{(t)}, \phi_i\big) \qquad (2)$$

Note that the above holds by the law of total probability for any choice of latent concept space.

To provide a more concrete example of latent concepts, consider the question-answering task in the BoolQ dataset. One of the questions in this dataset is "Did Abraham Lincoln write the letter in the film Saving Private Ryan?" to which the correct answer is "Yes". The latent concept, in this case, corresponds to the actual scene in the movie where the Bixby letter (written by Lincoln) is read to a group of soldiers. Just as in our case, first a concept $\theta$ is drawn, e.g., the film is made; then from the film, a string $\mathbf{x}$ is sampled, i.e., the previous question about the film.

For another example of latent concepts, we can think of arithmetic calculations such as multiplication. When we wish to express multiplication through language, we may write something like "$4 * 4$". The latent concepts behind this string are the mechanisms of multiplication (e.g., multiplication is just iterative addition, and addition itself is simply increasing the value of a number by one iteratively). These examples are intended to be easily digestible. However, latent concepts can also be significantly more abstract, such as a vector in some unknown embedding space.

## 4 A Theoretical Formulation of Multi-Agent Debate

We begin by providing a theoretical formulation of multi-agent debate. This formulation will provide key insights into the inner workings of the debate procedure, which we will use to improve debate.

The key behind our framework is to use the idea of latent concepts and expansion of each model's generation probability (Equation 2) in order to better understand debate. Prior to presenting our theoretical framework, we first state an important assumption.

**Assumption 4.1.** For a given latent concept space $\Theta$, the probability of generating response $\mathbf{z}_i^{(t+1)}$ is conditionally independent of both the responses $Z^{(t)}$ and the task $\mathbf{x}$, given concept $\boldsymbol{\theta} \in \Theta$ and model parameters $\phi_i$, i.e., $\mathbb{P}\big(\mathbf{z}_i^{(t+1)} \big| \boldsymbol{\theta}, \mathbf{x}, Z^{(t)}, \phi_i\big) = \mathbb{P}\big(\mathbf{z}_i^{(t+1)} \big| \boldsymbol{\theta}, \phi_i\big)$.

---

[1] In practice, $a$ can be a regular-expression checker or an LLM-based judge such as Liang et al. [2023]

Assumption 4.1 can be interpreted as saying that a model's generation $\mathbf{z}_i$, is uniquely determined by the model's parameters $\phi_i$ and the concepts identified by a model, namely $\boldsymbol{\theta}$. In the case of encoder-decoder-based models, one can conceptualize the joint between $\phi$ and $\boldsymbol{\theta}$ as corresponding to the embedding produced by the encoder. With this embedding in hand, the original input $(\mathbf{x}, Z^{(t)})$ no longer influences the model's output, rather the embedding and model parameters will uniquely determine the model's output.

Next, we derive the following lemma which will be useful in examining the way that model responses evolve debate rounds.

**Lemma 4.2.** *The generation of model $i$ at time $t + 1$ can be expressed as,*

$$\mathbb{P}\big(\mathbf{z}_i^{(t+1)}|\ Z^{(t)}, \mathbf{x}, \boldsymbol{\phi}_i\big) \propto \sum_{\boldsymbol{\theta} \in \Theta} \underbrace{\mathbb{P}\big(\mathbf{z}_i^{(t+1)}|\boldsymbol{\theta}, \boldsymbol{\phi}_i\big)\mathbb{P}\big(\mathbf{x}|\boldsymbol{\theta}, \boldsymbol{\phi}_i\big)\mathbb{P}\big(\boldsymbol{\theta}|\boldsymbol{\phi}_i\big)}_{\textit{generation without other agents}} \underbrace{\prod_{j=1}^{n} \mathbb{P}\big(\mathbf{z}_j^{(t)}|\ \boldsymbol{\theta}, \boldsymbol{\phi}_i\big)}_{\textit{skew caused by other agents}}$$

The significance of this lemma is that we are able to express the probability of generating a given response $\mathbf{z}_i^{(t+1)}$ *with* the other model responses $Z^{(t)}$ in terms of the probability of generating $\mathbf{z}_i^{(t+1)}$ *without* the other model responses and a skew term caused by those model responses. Note that,

$$\mathbb{P}\big(\mathbf{z}_i^{(t+1)}|\mathbf{x}, \boldsymbol{\phi}_i\big) \propto \sum_{\boldsymbol{\theta} \in \Theta} \mathbb{P}\big(\mathbf{z}_i^{(t+1)}|\boldsymbol{\theta}, \boldsymbol{\phi}_i\big)\mathbb{P}\big(\mathbf{x}|\boldsymbol{\theta}, \boldsymbol{\phi}_i\big)\mathbb{P}\big(\boldsymbol{\theta}|\boldsymbol{\phi}_i\big)$$

Thus, we can think of $\mathbb{P}\big(\mathbf{z}_i^{(t+1)}|\ Z^{(t)}, \mathbf{x}, \boldsymbol{\phi}_i\big)$ as a weighted version of $\mathbb{P}\big(\mathbf{z}_i^{(t+1)}|\ \mathbf{x}, \boldsymbol{\phi}_i\big)$, where the weights are given by the skew term $\prod_{j=1}^{n} \mathbb{P}\big(\mathbf{z}_j^{(t)}|\ \boldsymbol{\theta}, \boldsymbol{\phi}_i\big)$.

**Debate and In-Context Learning**   To help conceptualize the role of latent concepts in debate, we discuss the work of Xie et al. [2021], which uses Bayesian inference over latent concepts to understand in-context learning. There are natural connections between in-context learning and multi-agent debate. In-context learning works as follows: given a task $\mathbf{x}$ and a model $f$, select several examples of task-answers pairs $(\mathbf{x}_1, y_1) \ldots (\mathbf{x}_m, y_m)$ which are *similar* to $\mathbf{x}$. Then prompt the model $f$ for an answer to task $\mathbf{x}$, given examples $(\mathbf{x}_j, y_j)$, i.e. $\mathbf{z} = f\big(\mathbf{x}|\ (\mathbf{x}_1, y_1) \ldots (\mathbf{x}_m, y_m)\big)$. A key result of Xie et al. [2021] is that latent concepts in the examples $(\mathbf{x}_j, y_j)$, particularly concepts shared between many examples, influence the model's answer $\mathbf{z}$. For any concept where $\mathbb{P}\big(\boldsymbol{\theta}|\ (\mathbf{x}_1, y_1) \ldots (\mathbf{x}_m, y_m)\big)$ is large relative to other concepts (i.e., there is a shared concept $\boldsymbol{\theta}$ between the examples), the model is more likely to give response $\mathbf{z}$ which also share that concept. Model responses at the previous round $Z^{(t)}$ serve a similar function to the examples of in-context learning. The model's updated response at time $t + 1$, namely $\mathbf{z}_i^{(t)}$, is influenced by concepts shared between the responses in $Z^{(t)}$. The skew term in Lemma 4.2 provides a glimpse of how latent concepts conveyed by $Z^{(t)}$ will influence the generation of $\mathbf{z}_i^{(t)}$, namely that $\prod_{j=1}^{n} \mathbb{P}\big(\mathbf{z}_j^{(t)}|\ \boldsymbol{\theta}, \boldsymbol{\phi}_i\big)$ reweighs the model's generation.

## 4.1   Debate Objective

Through this perspective of debate we can more effectively design debate procedures by leveraging the concept space $\Theta$. To do this, we will first formulate debate as an optimization problem where the *skew term*, described in Lemma 4.2, corresponds to the optimization variables. For a given task $\mathbf{x}$ and answer $y$, each round of debate can be formulated as the following optimization problem.

$$\underset{Z^{(t)}}{\arg\max} \sum_{i=0}^{n} \mathbb{P}\big(a(\mathbf{z}_i^{(t+1)}) = y|\ Z^{(t)}, \boldsymbol{\phi}_i\big)$$

At time $t$ we aim to craft responses $Z^{(t)}$ such that they maximize the probability of providing the correct answer at the next time step. Expanding this objective over the latent concept space $\Theta$, yields

$$\underset{Z^{(t)}}{\arg\max} \sum_{i=1}^{n} \sum_{\boldsymbol{\theta} \in \Theta} \left( \mathbb{P}\big(a(\mathbf{z}^{(t+1)}) = y|\ \boldsymbol{\theta}, \boldsymbol{\phi}_i\big)\mathbb{P}\big(\boldsymbol{\theta}|\ \boldsymbol{\phi}_i\big)\mathbb{P}\big(\mathbf{x}|\ \boldsymbol{\theta}, \boldsymbol{\phi}_i\big) \prod_{j=1}^{n} \mathbb{P}\big(\mathbf{z}_j^{(t)}|\ \boldsymbol{\theta}, \boldsymbol{\phi}_i\big) \right) \quad (3)$$

The key challenges with directly optimizing this objective are: 1) the true concept $\boldsymbol{\theta}^*$ from which $\mathbf{x}$ and $y$ originate, as well as the relationship between $\mathbf{z}_j^{(t)}$ and the underlying concepts, is unknown, 2) the responses in $Z^{(t)})$ are natural language. However, will allow us to design several approaches within the concept space to better optimize debate. To motivate these approaches, we first need to make several observations about the debate procedure as a whole.

## 5 Debate Principals

In this section, we discuss factors that affect the efficacy of LLMs debate. In particular, we look at the role of information diversity, both in terms of the diversity of responses in $Z^{(t)}$ as well as diversity in model capabilities. We see that a lack of diversity in either aspect is detrimental to the debate process. Lastly, we study a particular type of homogeneity in debate, namely shared misconceptions in which a large portion of models all share a similar erroneous belief about the task $x$.

### 5.1 Information Diversity

We begin by examining how the debate procedure is affected by both the diversity of model abilities and the diversity of model responses. Homogeneity in either ability or responses will bias the debate procedure towards certain latent concepts.

**Similar Model Capabilities**   Suppose the debate process is conducted with only one type of model (in effect $n$ copies of the same model). That is, $\phi_i \equiv \phi$ for all $i \in [n]$. Then, as the number of agents increases, the debate procedure is more greatly impacted by the echo chamber effect, i.e., the probability that a round of debate results in a change to the most likely concept, perceived by agents, approaches 0. That is, a greater number of similar agents results in static debate dynamics, in essence defeating the purpose of debate.

**Theorem 5.1.** *Suppose all $n$ agents have identical configurations ($\phi_i \equiv \phi$ for all $i$). For round $t > 0$ let $\boldsymbol{\theta}'^{(t)} = \left( \arg \max_{\boldsymbol{\theta} \in \Theta} \mathbb{P}\left(\boldsymbol{\theta} \mid \mathbf{x}, Z^{(t)}, \boldsymbol{\phi}\right)\right)$ and $\boldsymbol{\theta}'^{(t+1)} = \left( \arg \max_{\boldsymbol{\theta} \in \Theta} \mathbb{P}\left(\boldsymbol{\theta} \mid \mathbf{x}, Z^{(t+1)}, \boldsymbol{\phi}\right)\right)$, i.e., $\boldsymbol{\theta}'^{(t)}$ and $\boldsymbol{\theta}'^{(t+1)}$ are the concepts most likely to be inferred by a model with parameters $\phi$ when given task $\mathbf{x}$ and responses $Z^{(t)}$, $Z^{(t+1)}$ respectively. Then $\mathbb{P}\left(\boldsymbol{\theta}'^{(t)} = \boldsymbol{\theta}'^{(t+1)})\right) \to 1$ as $n \to \infty$.*

We defer a full proof to the Supplement, Section A. Theorem 5.1 implies that when debate is conducted with multiple copies of the same model (or very similar models), increasing the number of models results in debate centering on a single (unchanging) concept, rather than a balanced distribution over multiple concepts.

**Similar Model Opinions**   Next, we examine how similar responses impact the collaboration process. At time $t$ suppose that there are $n$ responses $Z^{(t)}$ and at least $m$ of those responses are *similar*, i.e., there exists some concept $\boldsymbol{\theta}'$ such that $\boldsymbol{\theta}' = \arg \max_{\boldsymbol{\theta} \in \Theta} \mathbb{P}\left(\boldsymbol{\theta} \mid \mathbf{z}_j^{(t)}, \phi_i\right)$ for all $j \leq m$. That is, each of the $m$ responses has a shared "most likely" concept when viewed by a model with parameters $\phi_i$.

**Theorem 5.2.** *Suppose that $Z^{(t)}$ contains at least $m$ responses with the property that $\boldsymbol{\theta}' = \arg \max_{\boldsymbol{\theta} \in \Theta} \mathbb{P}\left(\mathbf{z}_i^{(t)} \mid \boldsymbol{\theta}, \phi_i\right)$. Then, as $m \to \infty$ the model's generation at the next round $(t+1)$ becomes uniquely determined by a single concept $\boldsymbol{\theta}'$ i.e. $\dfrac{\mathbb{P}\left(\mathbf{z}_{(i,1)}^{(t+1)} \mid Z^{(t)}, \mathbf{x}, \phi_i\right)}{\mathbb{P}\left(\mathbf{z}_{(i,2)}^{(t+1)} \mid Z^{(t)}, \mathbf{x}, \phi_i\right)} \to \dfrac{\mathbb{P}\left(\mathbf{z}_{(i,1)}^{(t+1)} \mid \boldsymbol{\theta}', \phi_i\right)}{\mathbb{P}\left(\mathbf{z}_{(i,2)}^{(t+1)} \mid \boldsymbol{\theta}', \phi_i\right)}$ for all response pairs $\mathbf{z}_{(i,1)}^{(t+1)}, \mathbf{z}_{(i,2)}^{(t+1)}$.*

We defer a full proof to the Supplement Section A. Theorem 5.2 indicates the susceptibility that LLM debate has towards tyranny of the majority. If a large number of models provide similar responses to a task $\mathbf{x}$, then those repeated answers will drown out the single provided by the other models' responses, as well as the task $\mathbf{x}$ itself. In Section 7 we demonstrate that this occurs in practice.

### 5.2 Shared Misconceptions

Next, we study a particular type of homogeneity in model capabilities and responses, namely shared misconceptions. When a common misconception is shared among the models, debate is less effective

and is likely to converge to erroneous concepts associated with the shared misconception. We now formalize the notion of misconceptions.

**Definition 5.3. (Misconception)**: For a given concept $\boldsymbol{\theta}^*$, a model with parameters $i$ is said to have a *misconception* regarding $\boldsymbol{\theta}^*$ if there exists another concept $\boldsymbol{\theta}'$ s.t.,

$$\mathbb{P}_{\mathbf{x} \sim D(\boldsymbol{\theta}^*)}\big[\mathbb{P}(\mathbf{x}|\,\boldsymbol{\theta}', \boldsymbol{\phi}_i) > \mathbb{P}(\mathbf{x}|\,\boldsymbol{\theta}^*, \boldsymbol{\phi}_i)\big] > 1/2$$

That is, for tasks generated from the concept $\boldsymbol{\theta}^*$, the model believes that the erroneous concept $\boldsymbol{\theta}'$ explains more than half of the tasks better than the true concept $\boldsymbol{\theta}^*$.

There is said to be a *shared misconception* if $m$ of agents have a misconception and share the same erroneous concept $\boldsymbol{\theta}'$. When the models share a common misconception the responses produced by those models are biased towards the erroneous concept $\boldsymbol{\theta}'$.

**Theorem 5.4.** *Let the true concept be $\boldsymbol{\theta}^*$ and suppose that $m$ of the $n$ agents have a shared misconception for erroneous concept $\boldsymbol{\theta}'$. Then, task and answer $(\mathbf{x}, y) \sim D(\boldsymbol{\theta}^*)$ expected average correctness of the debate procedure at the final round $T$ is monotonically decreasing with $m$, i.e., $\frac{1}{n} \sum_i^n \mathbb{P}\big(a(\mathbf{z}_i^{(T)}) = y\big)$ is decreasing with $m$.*

We defer the full proof to the supplement Section A. It should be noted that the phenomenon of converging to erroneous concepts is unlikely to be mitigated by adding more models. When the misconceptions of one model are formed through training data, it is likely that other models will possess the same misconception unless specifically trained to avoid such errors due to the high correlation in training data between models.

# 6 Interventions

In this section, we discuss several modifications to the debate procedure, referred to as *interventions*. We break our interventions into two categories: **Pruning** which focuses on choosing which responses to keep in $Z^{(t)}$, and **Modifying** which focuses on changing or editing the responses $Z^{(t)}$.

## 6.1 Pruning Interventions

At round $t$ of debate, running interventions work by selecting only a subset of responses $Z'^{(t)}$ from $Z^{(t)}$ before starting the next round $t + 1$. When using a pruning intervention, the models at round $t + 1$ will see only the pruned response set $Z'^{(t)}$, rather than the full response set $Z^{(t)}$.

**Diversity Pruning**   Let KL represent Kullback–Leibler divergence. The diversity pruning intervention selects $k$ of the $n$ responses in $Z^{(t)}$ which maximizes information entropy, i.e.,

$$Z'^{(t)} = \underset{Y \subset Y^{(t)}}{\operatorname{argmax}} \sum_{\mathbf{z}_i, \mathbf{z}_j \in Z} \mathrm{KL}\big(D(\boldsymbol{\theta}|\,\mathbf{z}_i),\ D(\boldsymbol{\theta}|\,\mathbf{z}_j)\big) \qquad \text{s.t. } |Z| = k$$

**Quality Pruning**   Quality pruning aims to select the $k$ responses in $Z^{(t)}$ with the highest similarity to the task $x$. Similar to diversity pruning, quality pruning selects $k$ of the $n$ responses at time $t$. Rather than selecting for diversity, quality pruning aims to select the $k$ highest question responses. This is done by selecting the $k$ responses which maximize

$$Z'^{(t)} = \underset{Z \subset Z^{(t)}}{\operatorname{argmin}} \sum_{\mathbf{z}_i, \in Z} \mathrm{KL}\big(D(\boldsymbol{\theta}|\,\mathbf{x}),\ D(\boldsymbol{\theta}|\,\mathbf{z}_i)\big) \qquad \text{s.t. } |Z| = k$$

In practice computing $\mathrm{KL}\big(D(\boldsymbol{\theta}|\mathbf{x}), D(\boldsymbol{\theta}|\mathbf{z}_i)\big)$ or $\mathrm{KL}\big(D(\boldsymbol{\theta}|\mathbf{z}_i), D(\boldsymbol{\theta}|\mathbf{z}_j)\big)$ is intractable. However, sentence embedding can be used as a proxy for these values. Section C discusses this in further detail.

Next, we show that when models have a shared misconception, diversity pruning decreases the likelihood that the debate procedure will converge to the erroneous concept corresponding to the shared misconception.

**Theorem 6.1.** *Let the true concept be $\boldsymbol{\theta}^*$ and suppose that at least $n/2$ agents have a shared misconception for erroneous concept $\boldsymbol{\theta}'$. Then diversity pruning decreases the probability that debate converges to an answer $y'$ which is sourced from the erroneous concept $\boldsymbol{\theta}'$, i.e. $y' \sim D(\boldsymbol{\theta}')$.*

We defer the full proof to the Supplement, Section A.

**Theorem 6.2.** *For a given task-answer pair* $(\mathbf{x}, y)$ *quality pruning increases the probability that debate converges to the correct answer, i.e. let* $Z^{(t)}$ *be the set of all responses at time* $t$ *and* $Z'^{(t)}$ *be the result of quality pruning, then* $\sum_{i=1}^{n} \mathbb{P}(a(\mathbf{z}_i^{(t+1)}) = y|\mathbf{x}, Z'^{(t)}, \boldsymbol{\phi}_i) > \sum_{i=1}^{n} \mathbb{P}(a(\mathbf{z}_i^{(t+1)} = y|\mathbf{x}, Z^{(t)}, \boldsymbol{\phi}_i).$

We defer the full proof to the Supplement, Section A.

**Remark 6.3.** As shown by Theorems 6.1 and 6.2, diversity pruning decreases the probability that debate converges to incorrect answers sourced from a particular concept, while quality pruning increases the probability that debate converges to a correct answer sourced from the true concept. Both interventions can be used simultaneously to guide the debate procedure more effectively away from wrong answers and towards correct answers.

## 6.2 Modification Interventions

**Misconception Refutation**    In addition to selecting which responses in $Z^{(t)}$ will be used in the next round of debate, we can also modify the responses in $Z^{(t)}$. Misconception refutation aims to do precisely this by updating response $\mathbf{z}_j^{(t)}$ to be more relevant to the task $\mathbf{x}$.

$$\mathbf{z}_j^* = \arg\min_{\mathbf{z}} \mathrm{KL}\big(D(\boldsymbol{\theta}|\mathbf{x}),\ D(\boldsymbol{\theta}|\,\mathbf{z})\big) - \mathrm{KL}\big(D(\boldsymbol{\theta}|\mathbf{z}_j^{(t)}),\ D(\boldsymbol{\theta}|\,\mathbf{z})\big)$$

Similar to Diversity Pruning and Quality Pruning, the distributions in the above objective are intractable in practice. As such, we use a proxy to update each response $\mathbf{z}_j^{(t)}$, specifically produce $\mathbf{z}_j^*$ by having an LLM minimally modify the given response $\mathbf{z}_j^{(t)}$. The model is first prompted for a list of misconceptions and errors identified in the response. Given the list of misconceptions, the model is asked for both a refutation of the misconception and a corrected version of the response. For more details, see Section C of the Supplement.

**Theorem 6.4.** *For task-answer pair* $(\mathbf{x}, y)$, *misconception refutation increases the probability of debate converging to the correct answer, i.e. let* $Z^{(t)}, Z^{*(t)}$ *be the responses before and after misconception refutation, then* $\sum_{i=1}^{n} \mathbb{P}(a(\mathbf{z}_i^{(t+1)}) = y|\mathbf{x}, Z^{*(t)}, \boldsymbol{\phi}_i) > \sum_{i=1}^{n} \mathbb{P}(a(\mathbf{z}_i^{(t+1)} = y|\mathbf{x}, Z^{(t)}, \boldsymbol{\phi}_i).$

## 7 Experiments

**Experimental Design**    We conduct experiments on four common language model benchmarks. **BoolQ** Clark et al. [2019], which consists of $3,270$ yes-no questions, **MMLU** Hendrycks et al. [2020] which consists of $13,869$ multiple-choice questions (we use the $3,406$ high-school-level questions), **TruthfulQA** Lin et al. [2021] which consists of $817$ open-ended questions, and **MathQ** which consists of $3,000$ arithmetic questions of the from $a{\cdot}b{\cdot}c + d{\cdot}e{\cdot}f$. In the BoolQ, MMLU, MathQ, datasets model correctness is measured through regular expression matching. In the TruthfulQA dataset, model correctness is measured via an LLM judge (we use GPT-4 as the judge in all experiments)

We use four LLMs of increasing capability, **GPT-3.5** (GPT-3.5 Turbo) OpenAI [2022], **Llama-2** (Llama-2 7B Chat) Touvron et al. [2023], **Llama-3** (Llama-3 8B Instruct) Meta AI [2024], and **Mistral** (Mistral 7B Instruct v0.2) Jiang et al. [2023]. For sentence embeddings (which serve as a proxy of the latent concepts $\Theta$), we use sentence embeddings from **ADA-2** OpenAI [2022]. We compare a combination of our three interventions **Ours** (see Algorithm 1 full details) with the debate paradigm of Du et al. [2023] (Society of Minds) **SoM**.

We begin by making several empirical observations about the multi-agent debate process.

**Tyranny of the Majority**    First, we examine the susceptibility of models towards agreement with the majority opinion. That is, how likely are models to give a specific answer at round $t + 1$ when $m$ of the models provided that specific answer at round the previous round (round $t$)? For example, in BoolQ suppose the specific answer is "Yes", then we want to know: how likely is a model to give a "Yes"-answer at round $t + 1$ if that model observes $m$ "Yes" answers at round $t$.

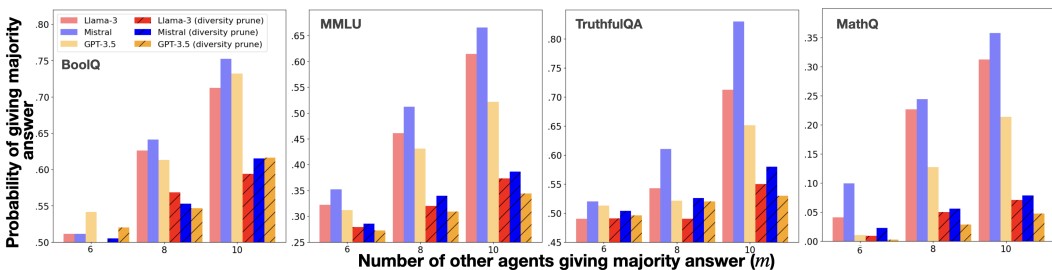

Figure 1: Probability that each model echoes the majority answer at round $t = 11$, as the number of responses at time $t = 0$ gives that majority answer (debate between 12 models are used).

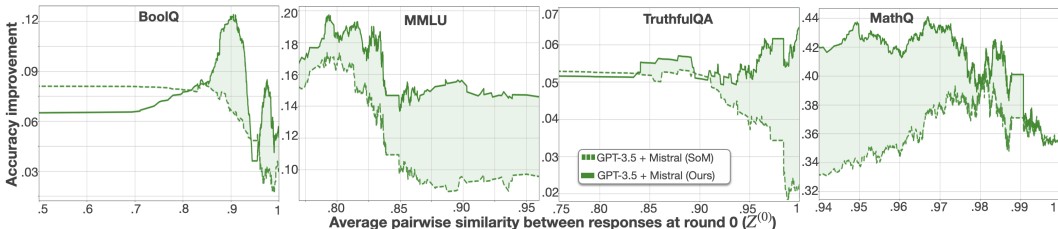

Figure 2: Average accuracy improvement as a function of response diversity at round $0$ of debate.

To measure this, we first select a random target answer, e.g., "Yes", and then prompt $m$ of the models (out of 11) to provide responses "Yes"-answers[2] (while the other $11$-$m$ models are prompted to provide a different randomly selected answer). These 11 responses make up $Z^{(t)}$, we then test each model's likelihood of providing the target at round $t+1$ when observing $Z^{(t)}$ before diversity pruning (solid) and after diversity running (hatched).

In Figure 1, we see models are susceptible to echo chamber effects (this phenomenon is predicted by Theorem 5.1). The likelihood of providing the majority answer increases when $Z^{(t)}$ contains more instances of the majority answer (i.e., as $m$ increases). Figure 1 also demonstrates that diversity pruning (with $k = 5$) reduces this echo chamber effect. See the Supplement for details.

**Diversity of Opinions** Next, we examine the effectiveness of SoM and our method as a function of opinion diversity. Figure 2 shows the average accuracy improvement of SoM (dashed) and our method (solid) over single model performance (i.e., average performance at round $t = 0$), as a function of the similarity between all responses at round $t = 0$ of debate (measured via pairwise cosine similarity). We see that for BoolQ, MMLU, and TruthfulQA, SoM is less effective when the similarity between responses increases. This observation is predicted by Theorems 5.1 and 5.2, which show that debate, without interventions, is less effective when model responses are too similar.

We see that our method's improvement compared to SoM is greatest when model opinions are more similar (cosine similarity close to 1). Note that the MathQ benchmark, where responses consist primarily of arithmetic, serves as a counter-example to these observations. This is due to the fact that sentence embedding of any two arithmetic expansions will be similar, regardless of their *true* similarity; as such, the cosine similarity between embedding is less meaningful on this benchmark.

**Debate Interventions** Now, we examine the effectiveness of a combination of our three debate interventions (see Algorithm 1 for full details of how the interventions are combined). We begin with a per-round performance of our method and SoM, as shown in Figure 3. We see that typically, the advantage of our method over debate arises in the later rounds of debate. Next, in Table 1, we present a full set of results for single models, SoM, and a combination of our three interventions. In all cases, our method is either competitive with, or superior to, SoM.

---

[2]Any model *not* providing the target answer is re-prompted until the target answer is provided.

| | Single | SoM | Ours | Single | SoM | Ours |
|---|---|---|---|---|---|---|
| | **BoolQ** | | | **MMLU** | | |
| $6\times$ GPT-3.5 | $.80_{\pm.014}$ | $.84_{\pm.012}$ | $.85_{\pm.012}$ | $.73_{\pm.014}$ | $.74_{\pm.016}$ | $\mathbf{.79}_{\pm\mathbf{.014}}$ |
| $6\times$ Llama-3 | $.76_{\pm.014}$ | $.78_{\pm.013}$ | $.78_{\pm.013}$ | $.67_{\pm.016}$ | $.70_{\pm.015}$ | $.75_{\pm.014}$ |
| $6\times$ Llama-2 | $.67_{\pm.017}$ | $.68_{\pm.017}$ | $.73_{\pm.016}$ | $.41_{\pm.017}$ | $.47_{\pm.018}$ | $.52_{\pm0.18}$ |
| $6\times$ Mistral | $.80_{\pm.014}$ | $.82_{\pm.013}$ | $.85_{\pm.012}$ | $.66_{\pm.016}$ | $.65_{\pm.016}$ | $.66_{\pm.016}$ |
| $3\times$ GPT-3.5 + $3\times$ Llama-3 | - | $.82_{\pm.014}$ | $.84_{\pm.013}$ | - | $.73_{\pm.016}$ | $.78_{\pm.016}$ |
| $3\times$ GPT-3.5 + $3\times$ Mistral | - | $.83_{\pm.013}$ | $\mathbf{.87}_{\pm.012}$ | - | $.69_{\pm.017}$ | $.72_{\pm.016}$ |
| $3\times$ Llama-3 + $3\times$ Mistral | - | $.80_{\pm.014}$ | $.80_{\pm.014}$ | - | $.69_{\pm.017}$ | $.74_{\pm.016}$ |
| | **TruthfulQA** | | | **Math** | | |
| $6\times$ GPT-3.5 | $.61_{\pm.033}$ | $.63_{\pm.032}$ | $\mathbf{.69}_{\pm.030}$ | $.53_{\pm.035}$ | $.88_{\pm.016}$ | $\mathbf{.93}_{\pm.01}$ |
| $6\times$ Llama-2 | $.47_{\pm.034}$ | $.52_{\pm0.35}$ | $.55_{\pm.034}$ | $.11_{\pm.013}$ | $.13_{\pm.014}$ | $.19_{\pm.015}$ |
| $6\times$ Llama-3 | $.53_{\pm.035}$ | $.55_{\pm.032}$ | $.55_{\pm.032}$ | $.25_{\pm.016}$ | $.33_{\pm.017}$ | $.48_{\pm.018}$ |
| $6\times$ Mistral | $.48_{\pm.034}$ | $.51_{\pm.035}$ | $.53_{\pm.034}$ | $.13_{\pm.013}$ | $.19_{\pm.014}$ | $.18_{\pm.014}$ |
| $3\times$ GPT-3.5 + $3\times$ Llama-3 | - | $.56_{\pm.035}$ | $.62_{\pm.031}$ | - | $.76_{\pm.015}$ | $.82_{\pm.014}$ |
| $3\times$ GPT-3.5 + $3\times$ Mistral | - | $.52_{\pm.035}$ | $.56_{\pm.035}$ | - | $.56_{\pm.018}$ | $.68_{\pm.017}$ |
| $3\times$ Llama-3 + $3\times$ Mistral | - | $.49_{\pm.036}$ | $.53_{\pm.035}$ | - | $.22_{\pm.015}$ | $.23_{\pm.015}$ |

Table 1: Accuracy of a solo model, debate, and our debate interventions: 10 rounds, 6 models.

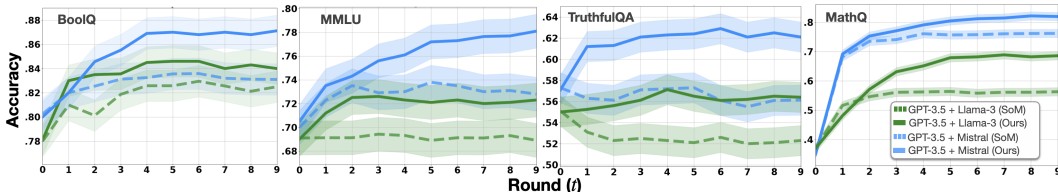

Figure 3: Accuracy per round, our method and SoM when combing GPT-3.5 with Llama-3 or Mistral.

In addition to providing results for the combination of our interventions, we also investigate the effectiveness of each intervention applied individually (see Table 3 of the Supplement). These results indicate that our method is most successful when applying all three interventions simultaneously. In fact, some interventions can be detrimental to the debate process when applied in isolation. This is expected as each intervention is inherently designed to be complementary.

## 8 Limitations

While we aim to address some of the fundamental issues of multi-LLM debate, such as tyranny of the majority, there are several factors that need to be considered when adopting our framework. Firstly, our theoretical results leverage a latent concept space, which may not be accessible in practice, necessitating the use of proxies such as sentence embeddings. Reliance on proxies is particularly consequential for quality and diversity pruning; these interventions are less effective in domains where sentence embeddings are less meaningful, e.g., arithmetic questions. Additionally, our interventions can increase the inference time of the debate procedure. Increased inference time stems primarily from misconception refutation, as this intervention requires re-prompting each debater multiple times.

## 9 Conclusion

Multi-agent debate is an effective tool for improving the efficacy of LLM responses. However, debate is naturally susceptible to issues such as tyranny of the majority and shared misconceptions between models. By making use of our theoretical framework for debate, we are able to establish interventions for the debate procedure which help to alleviate these issues and improve the general performance of multi-agent debate. We saw that diversity pruning reduces the influence of similar responses. This is especially helpful in settings where the majority of agents provide incorrect responses that share a common error. A combination of all three interventions consistently leads to better debate.

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

# Appendix

## A Theoretical Results

*Proof of Lemma 4.2.* This result holds via marginalization of the posterior predictive distribution over the latent concepts $\Theta$, namely

$$\mathbb{P}\big(\mathbf{z}_i^{(t+1)}|\ Z^{(t)}, \mathbf{x}, \boldsymbol{\phi}_i\big) = \sum_{\boldsymbol{\theta} \in \Theta} \mathbb{P}\big(\mathbf{z}_i^{(t+1)}|\ \boldsymbol{\theta}, Z^{(t)}, \mathbf{x}, \boldsymbol{\phi}_i\big) \mathbb{P}\big(\boldsymbol{\theta}|\ Z^{(t)}, \mathbf{x}, \boldsymbol{\phi}_i\big)$$

$$= \sum_{\boldsymbol{\theta} \in \Theta} \mathbb{P}\big(\mathbf{z}_i^{(t+1)}|\ \boldsymbol{\theta}, \boldsymbol{\phi}_i\big) \mathbb{P}\big(Z^{(t)}, \mathbf{x}|\boldsymbol{\theta}, \boldsymbol{\phi}_i\big) \frac{\mathbb{P}\big(\boldsymbol{\theta}|\ \boldsymbol{\phi}_i\big)}{\mathbb{P}\big(Z^{(t)}, \mathbf{x}\big)|\boldsymbol{\phi}_i\big)}$$

$$\propto \sum_{\boldsymbol{\theta} \in \Theta} \mathbb{P}(\mathbf{z}_i^{(t+1)}|\ \boldsymbol{\theta}, \boldsymbol{\phi}_i) \mathbb{P}(\mathbf{x}|\ \boldsymbol{\theta}, \boldsymbol{\phi}_i) \mathbb{P}(\boldsymbol{\theta}|\ \boldsymbol{\phi}_i) \prod_{j=1}^{n} \mathbb{P}\big(\mathbf{z}_j^{(t)}|\ \boldsymbol{\theta}, \boldsymbol{\phi}_i\big)$$

$\square$

*Proof of Theorem 5.1.* Since each model has an identical configuration, i.e. $\boldsymbol{\phi}_i = \boldsymbol{\phi}_j$ for all $i, j \in [n]$, we simply refer to the configuration as $\boldsymbol{\phi}$. For the given task $\mathbf{x}$ let $\boldsymbol{\theta}^{*(t)}$ be the realization of $\boldsymbol{\theta}$ at time step $t$. Let $Z^{(t)} = \big(\mathbf{z}_1^{(t)}, \dots, \mathbf{z}_n^{(t)}\big)$, where each $\mathbf{z}_j^{(t)} \sim q(\mathbf{z}|\ \mathbf{x}, Z^{(t)}, \boldsymbol{\phi})$. Then the conditional density for each concept $\boldsymbol{\theta}$ can be written as,

$$\mathbb{P}\big(\boldsymbol{\theta}|\ Z^{(t)}, \mathbf{x}, \boldsymbol{\phi}\big)$$
$$\propto \mathbb{P}\big(Z^{(t)}, \mathbf{x}|\ \boldsymbol{\theta}, \boldsymbol{\phi}\big) \mathbb{P}\big(\boldsymbol{\theta}|\ \boldsymbol{\phi}\big)$$
$$= \mathbb{P}\big(\mathbf{x}|\ \boldsymbol{\theta}, \boldsymbol{\phi}\big) \mathbb{P}\big(\boldsymbol{\theta}|\ \boldsymbol{\phi}\big) \prod_{j=1}^{n} \mathbb{P}\big(\mathbf{z}_j^{(t)}|\ \boldsymbol{\theta}, \boldsymbol{\phi}\big)$$

where the term $\mathbb{P}\big(\mathbf{x}|\ \boldsymbol{\theta}, \boldsymbol{\phi}\big)\big(\theta|\ \boldsymbol{\phi}\big)$ is a constant with respect to the number of agents $n$. Since $\mathbb{P}\big(\boldsymbol{\theta}|\boldsymbol{\phi}\big) > 0$ for all $\boldsymbol{\theta} \in \theta$, each $\mathbf{z}_j^{(t)}$ is an i.i.d. draw from $q(\mathbf{z}|\ \mathbf{x}, Z^{(t-1)}, \boldsymbol{\phi})$, then

$$\boldsymbol{\theta}^{*(t)} = \lim_{n \to \infty} \left( \arg\max_{\boldsymbol{\theta} \in \Theta} \mathbb{P}\big(\boldsymbol{\theta}|\ Z^{(t)}, \mathbf{x}, \boldsymbol{\phi}\big) \right)$$

Thus, as $n \to \infty$, all models predict the same concept, namely $\boldsymbol{\theta}^{*(t)}$, at timestep $t$ with probability 1. $\square$

*Proof of Theorem 5.2.* Consider any two responses from agent $i$ at time $t + 1$, namely $\mathbf{z}_{(i,1)}^{(t+1)}, \mathbf{z}_{(i,2)}^{(t+1)}$. When there are $n$ duplicate messages, the ratio between the conditional generation probabilities of both responses can be written as

$$\frac{\mathbb{P}\big(\mathbf{z}_{(i,1)}^{(t+1)}|\ Z^{(t)}, \mathbf{x}, \boldsymbol{\phi}_i\big)}{\mathbb{P}\big(\mathbf{z}_{(i,2)}^{(t+1)}|\ Z^{(t)}, \mathbf{x}, \boldsymbol{\phi}_i\big)}$$

$$= \frac{\sum_{\boldsymbol{\theta} \in \Theta} \mathbb{P}\big(\mathbf{z}_{(i,1)}^{(t+1)}|\ \boldsymbol{\theta}, \boldsymbol{\phi}_i\big) \mathbb{P}\big(\boldsymbol{\theta}|\ Z^{(t)}, \mathbf{x}, \boldsymbol{\phi}_i\big)}{\sum_{\boldsymbol{\theta} \in \Theta} \mathbb{P}\big(\mathbf{z}_{(i,2)}^{(t+1)}|\ \boldsymbol{\theta}, \boldsymbol{\phi}_i\big) \mathbb{P}\big(\boldsymbol{\theta}|\ Z^{(t)}, \mathbf{x}, \boldsymbol{\phi}_i\big)}$$

$$= \frac{\sum_{\boldsymbol{\theta} \in \Theta} \mathbb{P}\big(\mathbf{z}_{(i,1)}^{(t+1)}|\ \boldsymbol{\theta}, \boldsymbol{\phi}_i\big) \mathbb{P}\big(Z^{(t)}, \mathbf{x}|\ \boldsymbol{\theta}, \boldsymbol{\phi}_i\big) \mathbb{P}\big(\boldsymbol{\theta}|\ \boldsymbol{\phi}_i\big)}{\sum_{\boldsymbol{\theta} \in \Theta} \mathbb{P}\big(\mathbf{z}_{(i,2)}^{(t+1)}|\ \boldsymbol{\theta}, \boldsymbol{\phi}_i\big) \mathbb{P}\big(Z^{(t)}, \mathbf{x}|\ \boldsymbol{\theta}, \boldsymbol{\phi}_i\big) \mathbb{P}\big(\boldsymbol{\theta}|\ \boldsymbol{\phi}_i\big)} \qquad \text{(factoring out common divisor } \mathbb{P}\big(Z^{(t)}, \mathbf{x}|\ \boldsymbol{\phi}_i\big)\text{)}$$

$$= \frac{\sum_{\boldsymbol{\theta} \in \Theta} \mathbb{P}\big(\mathbf{z}_{(i,1)}^{(t+1)}|\ \boldsymbol{\theta}, \boldsymbol{\phi}_i\big) \mathbb{P}(\mathbf{x}|\ \boldsymbol{\theta}, \boldsymbol{\phi}_i) \mathbb{P}(\boldsymbol{\theta}|\ \boldsymbol{\phi}_i) \Big( \prod_{j=1}^{k} \mathbb{P}\big(\mathbf{z}_j^{(t)}|\ \boldsymbol{\theta}, \boldsymbol{\phi}_i\big) \Big) \Big( \mathbb{P}\big(\mathbf{z}'^{(t)}|\ \boldsymbol{\theta}, \boldsymbol{\phi}_i\big) \Big)^m}{\sum_{\boldsymbol{\theta} \in \Theta} \mathbb{P}\big(\mathbf{z}_{(i,2)}^{(t+1)}|\ \boldsymbol{\theta}, \boldsymbol{\phi}_i\big) \mathbb{P}(\mathbf{x}|\ \boldsymbol{\theta}, \boldsymbol{\phi}_i) \mathbb{P}(\boldsymbol{\theta}|\ \boldsymbol{\phi}_i) \Big( \prod_{j=1}^{k} \mathbb{P}\big(\mathbf{z}_j^{(t)}|\ \boldsymbol{\theta}, \boldsymbol{\phi}_i\big) \Big) \Big( \mathbb{P}\big(\mathbf{z}'^{(t)}|\ \boldsymbol{\theta}, \boldsymbol{\phi}_i\big) \Big)^m}$$

Let $\boldsymbol{\theta}'$ be the concept which model $i$, with configuration $\phi_i$, believes is most likely to have produced response $\mathbf{z}'$, i.e.,

$$\boldsymbol{\theta}' = \arg\max_{\boldsymbol{\theta}} \mathbb{P}\big(\mathbf{z}'|\,\boldsymbol{\theta},\phi_i\big)$$

Then the above ratio can be rewritten as

$$
\frac{\sum_{\boldsymbol{\theta}\in\Theta} \mathbb{P}\big(\mathbf{z}^{(t+1)}_{(i,1)}|\,\boldsymbol{\theta},\phi_i\big)\mathbb{P}(\mathbf{x}|\,\boldsymbol{\theta},\phi_i)\mathbb{P}(\boldsymbol{\theta}|\,\phi_i)\Big(\prod_{j=1}^{k}\mathbb{P}\big(\mathbf{z}^{(t)}_j|\,\boldsymbol{\theta},\phi_i\big)\Big)\Big(\mathbb{P}\big(\mathbf{z}'^{(t)}|\,\boldsymbol{\theta},\phi_i\big)\Big)^{m}\overline{\Big(\mathbb{P}\big(\mathbf{z}'^{(t)}|\,\boldsymbol{\theta}',\phi_i\big)\Big)^{m}}^{\frac{1}{}}}{\sum_{\boldsymbol{\theta}\in\Theta} \mathbb{P}\big(\mathbf{z}^{(t+1)}_{(i,2)}|\,\boldsymbol{\theta},\phi_i\big)\mathbb{P}(\mathbf{x}|\,\boldsymbol{\theta},\phi_i)\mathbb{P}(\boldsymbol{\theta}|\,\phi_i)\Big(\prod_{j=1}^{k}\mathbb{P}\big(\mathbf{z}^{(t)}_j|\,\boldsymbol{\theta},\phi_i\big)\Big)\Big(\mathbb{P}\big(\mathbf{z}'^{(t)}|\,\boldsymbol{\theta},\phi_i\big)\Big)^{m}\overline{\Big(\mathbb{P}\big(\mathbf{z}'^{(t)}|\,\boldsymbol{\theta}',\phi_i\big)\Big)^{m}}^{\frac{1}{}}}
$$

$$
= \frac{\sum_{\boldsymbol{\theta}\in\Theta:\boldsymbol{\theta}\neq\boldsymbol{\theta}'} \mathbb{P}\big(\mathbf{z}^{(t+1)}_{(i,1)}|\,\boldsymbol{\theta},\phi_i\big)\mathbb{P}(\mathbf{x}|\,\boldsymbol{\theta},\phi_i)\mathbb{P}(\boldsymbol{\theta}|\,\phi_i)\Big(\prod_{j=1}^{k}\mathbb{P}\big(\mathbf{z}^{(t)}_j|\,\boldsymbol{\theta},\phi_i\big)\Big)\Big(\frac{\mathbb{P}\big(\mathbf{z}'^{(t)}|\,\boldsymbol{\theta},\phi_i\big)}{\mathbb{P}\big(\mathbf{z}'^{(t)}|\,\boldsymbol{\theta}',\phi_i\big)}\Big)^{m}}{\sum_{\boldsymbol{\theta}\in\Theta:\boldsymbol{\theta}\neq\boldsymbol{\theta}'} \mathbb{P}\big(\mathbf{z}^{(t+1)}_{(i,2)}|\,\boldsymbol{\theta},\phi_i\big)\mathbb{P}(\mathbf{x}|\,\boldsymbol{\theta},\phi_i)\mathbb{P}(\boldsymbol{\theta}|\,\phi_i)\Big(\prod_{j=1}^{k}\mathbb{P}\big(\mathbf{z}^{(t)}_j|\,\boldsymbol{\theta},\phi_i\big)\Big)\Big(\frac{\mathbb{P}\big(\mathbf{z}'^{(t)}|\,\boldsymbol{\theta},\phi_i\big)}{\mathbb{P}\big(\mathbf{z}'^{(t)}|\,\boldsymbol{\theta}',\phi_i\big)}\Big)^{m}}
$$

$$
\cdot\; \frac{+\mathbb{P}\big(\mathbf{z}^{(t+1)}_{(i,2)}|\,\boldsymbol{\theta}',\phi_i\big)\mathbb{P}(\mathbf{x}|\,\boldsymbol{\theta}',\phi_i)\mathbb{P}(\boldsymbol{\theta}'|\,\phi_i)\Big(\prod_{j=1}^{k}\mathbb{P}\big(\mathbf{z}^{(t)}_j|\,\boldsymbol{\theta}',\phi_i\big)\Big)}{+\mathbb{P}\big(\mathbf{z}^{(t+1)}_{(i,2)}|\,\boldsymbol{\theta}',\phi_i\big)\mathbb{P}(\mathbf{x}|\,\boldsymbol{\theta}',\phi_i)\mathbb{P}(\boldsymbol{\theta}'|\,\phi_i)\Big(\prod_{j=1}^{k}\mathbb{P}\big(\mathbf{z}^{(t)}_j|\,\boldsymbol{\theta}',\phi_i\big)\Big)}
$$

For each $\boldsymbol{\theta}\neq\boldsymbol{\theta}'$ we have $\frac{\mathbb{P}\big(\mathbf{z}'|\,\boldsymbol{\theta},\phi_i\big)}{\mathbb{P}\big(\mathbf{z}'|\,\boldsymbol{\theta}',\phi_i\big)} < 1$. Therefore

$$\lim_{m\to\infty}\left(\frac{\mathbb{P}\big(\mathbf{z}'|\,\boldsymbol{\theta},\phi_i\big)}{\mathbb{P}\big(\mathbf{z}'|\,\boldsymbol{\theta}',\phi_i\big)}\right)^{m} = 0$$

Note that the only summands which do not have an exponential dependency on $m$ are those associated with concept $\boldsymbol{\theta}'$. Therefore, when examining the limit of the above ratio with respect to the number of repeated signals $m$, we get

$$
\lim_{m\to\infty} \frac{\sum_{\boldsymbol{\theta}\in\Theta} \mathbb{P}\big(\mathbf{z}^{(t+1)}_{(i,1)}|\,\boldsymbol{\theta},\phi_i\big)\mathbb{P}(\mathbf{x}|\,\boldsymbol{\theta},\phi_i)\mathbb{P}(\boldsymbol{\theta}|\,\phi_i)\Big(\prod_{j=1}^{k}\mathbb{P}\big(\mathbf{z}^{(t)}_j|\,\boldsymbol{\theta},\phi_i\big)\Big)\Big(\mathbb{P}\big(\mathbf{z}'^{(t)}|\,\boldsymbol{\theta},\phi_i\big)\Big)^{m}}{\sum_{\boldsymbol{\theta}\in\Theta} \mathbb{P}\big(\mathbf{z}^{(t+1)}_{(i,2)}|\,\boldsymbol{\theta},\phi_i\big)\mathbb{P}(\mathbf{x}|\,\boldsymbol{\theta},\phi_i)\mathbb{P}(\boldsymbol{\theta}|\,\phi_i)\Big(\prod_{j=1}^{k}\mathbb{P}\big(\mathbf{z}^{(t)}_j|\,\boldsymbol{\theta},\phi_i\big)\Big)\Big(\mathbb{P}\big(\mathbf{z}'^{(t)}|\,\boldsymbol{\theta},\phi_i\big)\Big)^{m}}
$$

$$
= \frac{\mathbb{P}\big(\mathbf{z}^{(t+1)}_{(i,1)}|\,\boldsymbol{\theta}',\phi_i\big)\mathbb{P}(\mathbf{x}|\,\boldsymbol{\theta},\phi_i)\mathbb{P}(\boldsymbol{\theta}'|\,\phi_i)}{\mathbb{P}\big(\mathbf{z}^{(t+1)}_{(i,2)}|\,\boldsymbol{\theta}',\phi_i\big)\mathbb{P}(\mathbf{x}|\,\boldsymbol{\theta},\phi_i)\mathbb{P}(\boldsymbol{\theta}'|\,\phi_i)}
$$

$$
= \frac{\mathbb{P}\big(\mathbf{z}^{(t+1)}_{(i,1)}|\,\boldsymbol{\theta}',\phi_i\big)}{\mathbb{P}\big(\mathbf{z}^{(t+1)}_{(i,2)}|\,\boldsymbol{\theta}',\phi_i\big)}
$$

Thus the relationship between any two conditional generation probabilities can be uniquely defined by $\boldsymbol{\theta}'$. $\qquad\square$

*Proof of Theorem 5.4.* First, we examine the models' outputs at the first rounds of debate. For a model $i$ which possess the shared misconception, their conditional generation probability on the first round of debate can be expressed as,

$$\mathbb{P}\big(\mathbf{z}^{(t+1)}_i|\mathbf{x},\phi_i\big) \propto \sum_{\boldsymbol{\theta}} \mathbb{P}\big(\mathbf{z}^{(t+1)}_i|\,\boldsymbol{\theta},\phi_i\big)\mathbb{P}(\mathbf{x}|\,\boldsymbol{\theta},\phi_i)$$

Next, consider the term $\mathbb{P}\big(\mathbf{x}|\,\boldsymbol{\theta},\phi_i\big)$. Let $\phi_i'$ be a set of model parameters which does posses the common misconception. Then,

$$\mathbb{P}\big(\mathbf{x}|\,\boldsymbol{\theta},\phi_i\big) = \mathbb{P}\big(\mathbf{x},\mathbf{x}'|\,\boldsymbol{\theta},\phi_i'\big)$$

where $\mathbf{x}'$ is a message which conveys the erroneous concept $\boldsymbol{\theta}'$. Using this change of model parameters, we can express the condition generation probability as,

$$
\mathbb{P}\big(\mathbf{z}_i^{(t+1)}|\mathbf{x},\phi_i\big)
$$
$$
\propto \sum_{\boldsymbol{\theta}} \mathbb{P}\big(\mathbf{z}_i^{(t+1)}|\,\boldsymbol{\theta},\phi_i\big)\mathbb{P}\big(\mathbf{x},\mathbf{x}'|\,\boldsymbol{\theta},\phi_i'\big)
$$
$$
= \sum_{\boldsymbol{\theta}} \mathbb{P}\big(\mathbf{z}_i^{(t+1)}|\,\boldsymbol{\theta},\phi_i\big)\mathbb{P}\big(\mathbf{x}|\,\boldsymbol{\theta},\phi_i'\big)\mathbb{P}\big(\mathbf{x}'|\,\boldsymbol{\theta},\phi_i'\big)
$$

With this formulation of the conditional generation probability, we can the ratio between the true concept $\boldsymbol{\theta}^*$ and the erroneous concept $\boldsymbol{\theta}'$.

$$
\frac{\mathbb{P}\big(\mathbf{z}_i^{(t+1)}|\,\boldsymbol{\theta}^*,\phi_i\big)\mathbb{P}\big(\mathbf{x}|\,\boldsymbol{\theta}^*,\phi_i'\big)\mathbb{P}\big(\mathbf{x}'|\,\boldsymbol{\theta}^*,\phi_i'\big)}{\mathbb{P}\big(\mathbf{z}_i^{(t+1)}|\,\boldsymbol{\theta}',\phi_i\big)\mathbb{P}\big(\mathbf{x}|\,\boldsymbol{\theta}',\phi_i'\big)\mathbb{P}\big(\mathbf{x}'|\,\boldsymbol{\theta}',\phi_i'\big)}
$$
$$
< \frac{\mathbb{P}\big(\mathbf{z}_i^{(t+1)}|\,\boldsymbol{\theta}^*,\phi_i\big)\mathbb{P}\big(\mathbf{x}|\,\boldsymbol{\theta}^*,\phi_i'\big)}{\mathbb{P}\big(\mathbf{z}_i^{(t+1)}|\,\boldsymbol{\theta}',\phi_i\big)\mathbb{P}\big(\mathbf{x}|\,\boldsymbol{\theta}',\phi_i'\big)} \tag{4}
$$

which follows directly from $\mathbb{P}\big(\mathbf{x}'|\,\boldsymbol{\theta}',\phi_i'\big) > \mathbb{P}\big(\mathbf{x}'|\,\boldsymbol{\theta}^*,\phi_i'\big)$. When models have a shared misconception (left side of Equation 4),

$$
\mathbb{P}\big(\mathbf{z}_i^{(t+1)}|\mathbf{x},\phi_i\big) < \mathbb{P}\big(\mathbf{z}_i^{(t+1)}|\mathbf{x},\phi_i'\big)
$$

for any $\mathbf{z}_i$ with

$$
\mathbb{P}\big(\mathbf{z}_i^{(t+1)}|\,\boldsymbol{\theta}^*,\phi_i\big) < \mathbb{P}\big(\mathbf{z}_i^{(t+1)}|\,\boldsymbol{\theta}',\phi_i\big)
$$

Therefore, at timestep $t = 0$, models with the shared misconception are more likely to yield responses which correlate with $\boldsymbol{\theta}'$.

For rounds $t > 0$, we can express the conditional probability as,

$$
\mathbb{P}\big(\mathbf{z}_i^{(t+1)}|\,\mathbf{x},Z^{(t)},\phi_i\big)
$$
$$
\propto \sum_{\boldsymbol{\theta}} \mathbb{P}\big(\mathbf{z}_i^{(t+1)}|\,\boldsymbol{\theta},\phi_i\big) \prod_{j=1}^{n} \mathbb{P}\big(\mathbf{z}_j^{(t)}|\,\boldsymbol{\theta},\phi_i\big)\mathbb{P}\big(\mathbf{x},\mathbf{x}'|\,\boldsymbol{\theta},\phi_i'\big)
$$
$$
= \sum_{\boldsymbol{\theta}} \bigg( \mathbb{P}\big(\mathbf{z}_i^{(t+1)}|\,\boldsymbol{\theta},\phi_i\big) \prod_{j\leq m}^{n} \mathbb{P}\big(\mathbf{z}_j^{(t)}|\,\boldsymbol{\theta},\phi_i\big)
$$
$$
\prod_{j>m}^{n} \mathbb{P}\big(\mathbf{z}_j^{(t)}|\,\boldsymbol{\theta},\phi_i\big)\mathbb{P}\big(\mathbf{x}|\,\boldsymbol{\theta},\phi_i'\big)\mathbb{P}\big(\mathbf{x}'|\,\boldsymbol{\theta},\phi_i'\big) \bigg)
$$

For a given $\mathbf{x}$, the terms

$$
\prod_{j\leq m}^{n} \mathbb{P}\big(\mathbf{z}_j^{(t)}|\,\boldsymbol{\theta},\phi_i\big)\big(\mathbf{x}'|\,\boldsymbol{\theta},\phi_i'\big)
$$

are maximized, in expectation, for $\boldsymbol{\theta} = \boldsymbol{\theta}'$. Moreover, for the any concept $\theta \neq \boldsymbol{\theta}'$, the ratio

$$
\frac{\prod_{j\leq m}^{n} \mathbb{P}\big(\mathbf{z}_j^{(t)}|\,\boldsymbol{\theta}',\phi_i\big)\big(\mathbf{x}'|\,\boldsymbol{\theta}',\phi_i'\big)}{\prod_{j\leq m}^{n} \mathbb{P}\big(\mathbf{z}_j^{(t)}|\,\boldsymbol{\theta},\phi_i\big)\big(\mathbf{x}'|\,\boldsymbol{\theta},\phi_i'\big)}
$$

is monotonically increasing as $m$ increases. Therefore, for any round, a model with the shared misconception is more likely to generate answers correlating with $\boldsymbol{\theta}'$, than those without the shared misconception. Further, the likelihood of generating such answers increases with $m$. $\square$

*Proof of Theorem 6.1.* As shown in the proof of Theorem 5.4, we can express each model's conditional generation probability as

$$\mathbb{P}\big(\mathbf{z}_i^{(t+1)}\,|\,\mathbf{x}, Z^{(t)}, \phi_i\big)$$

$$\propto \sum_{\boldsymbol{\theta}} \mathbb{P}\big(\mathbf{z}_i^{(t+1)}\,|\,\boldsymbol{\theta}, \phi_i\big) \prod_{j=1}^{n} \mathbb{P}\big(\mathbf{z}_j^{(t)}\,|\,\boldsymbol{\theta}, \phi_i\big) \mathbb{P}\big(\mathbf{x}, \mathbf{x}'\,|\,\boldsymbol{\theta}, \phi_i'\big)$$

$$= \sum_{\boldsymbol{\theta}} \Bigg( \mathbb{P}\big(\mathbf{z}_i^{(t+1)}\,|\,\boldsymbol{\theta}, \phi_i\big) \prod_{j \leq m}^{n} \mathbb{P}\big(\mathbf{z}_j^{(t)}\,|\,\boldsymbol{\theta}, \phi_i\big)$$

$$\prod_{j > m}^{n} \mathbb{P}\big(\mathbf{z}_j^{(t)}\,|\,\boldsymbol{\theta}, \phi_i\big) \mathbb{P}\big(\mathbf{x}\,|\,\boldsymbol{\theta}, \phi_i'\big) \mathbb{P}\big(\mathbf{x}'\,|\,\boldsymbol{\theta}, \phi_i'\big) \Bigg)$$

where $\mathbf{x}'$ is a response which conveys the erroneous concept $\boldsymbol{\theta}'$. Under this expression, each response $\mathbf{z}_i^{(t)}$ is generated according to $D(\mathbf{x}, Z^{(t)}, \phi_i)$ where each distribution differs only by the model parameters $\phi_i$. For $i \leq m$, the model parameters $\phi_i$ posses the common misconception, i.e., each distribution $D(\mathbf{z}_j^{(t)}\,|\,\mathbf{x}, Z^{(t)}, \phi_i)$ has a common scaling factor $\mathbb{P}\big(\mathbf{x}'\,|\,\boldsymbol{\theta}, \phi_i'\big)$ which is maximized at $\boldsymbol{\theta} = \boldsymbol{\theta}'$. Since only these models share this scaling factor,

$$\mathbb{E}\big[\mathrm{KL}\big(D(\boldsymbol{\theta}'\,|\,\mathbf{z}_{i_1}),\ D(\boldsymbol{\theta}'\,|\,\mathbf{z}_{i_2})\big)\big] \geq \mathbb{E}\big[\mathrm{KL}\big(D(\boldsymbol{\theta}'\,|\,\mathbf{z}_{i_1}),\ D(\boldsymbol{\theta}'\,|\,\mathbf{z}_j)\big)\big]$$

whenever $i_1, i_2 \leq m < j$. Hence, diversity pruning is more likely to select terms from agents with $m < j$ (i.e., those without the shared misconception), then agents with $j \leq m$. Since this holds true on every round of debate, all responses in the debate process place a lower weight on responses associated with $\boldsymbol{\theta}'$, i.e., a lower weight is placed on responses which have a higher chance of being incorrect. □

*Proof of Theorem 6.2.* At each round $t$, quality pruning selection a set of $k$ responses,

$$Z'^{(t)} = \underset{Y \subset Z^{(t)}}{\mathrm{argmin}} \sum_{\mathbf{z}_i, \in Y} \mathrm{KL}\big(D(\boldsymbol{\theta}\,|\,\mathbf{x}),\ D(\boldsymbol{\theta}\,|\,\mathbf{z}_i)\big)$$

$$\text{s.t. } |Y| = k$$

As such, for any set of responses $Z^{(t)}$

$$\sum_{\mathbf{z}_i' \in Z'^{(t)}} \mathrm{KL}\big(D(\boldsymbol{\theta}\,|\,\mathbf{x}),\ D(\boldsymbol{\theta}\,|\,\mathbf{z}_i')\big) \leq \sum_{\mathbf{z}_i \in Z^{(t)}} \mathrm{KL}\big(D(\boldsymbol{\theta}\,|\,\mathbf{x}),\ D(\boldsymbol{\theta}\,|\,\mathbf{z}_i)\big)$$

Using the fact that $Z'^{(t)} \subset Z^{(t)}$, we can write,

$$\sum_{\mathbf{z}_i' \in Z'^{(t)}} \mathrm{KL}\big(D(\boldsymbol{\theta}\,|\,\mathbf{x}),\ D(\boldsymbol{\theta}\,|\,\mathbf{z}_i')\big)$$

$$\leq \sum_{\mathbf{z}_i' \in Z'^{(t)}} \mathrm{KL}\big(D(\boldsymbol{\theta}\,|\,\mathbf{x}),\ D(\boldsymbol{\theta}\,|\,\mathbf{z}_i')\big) + \sum_{\mathbf{z}_i \in Z^{(t)} \setminus Z'^{(t)}} \mathrm{KL}\big(D(\boldsymbol{\theta}\,|\,\mathbf{x}),\ D(\boldsymbol{\theta}\,|\,\mathbf{z}_i)\big)$$

$$\implies$$

$$0 \leq \sum_{\mathbf{z}_i \in Z^{(t)} \setminus Z'^{(t)}} \mathrm{KL}\big(D(\boldsymbol{\theta}\,|\,\mathbf{x}),\ D(\boldsymbol{\theta}\,|\,\mathbf{z}_i)\big)$$

Thus, for a random task and answer pair $(\mathbf{x}, y) \sim D(\boldsymbol{\theta}^*)$, the relationship between the correctness of answers in $Z^{(t)}$ and $Z'^{(t)}$ is

$$\mathbb{E}_{(\mathbf{x},y) \sim D(\boldsymbol{\theta}^*)}\Bigg[\frac{1}{|Z'^{(t)}|} \sum_{\mathbf{z}_i \in Z'^{(t)}} \mathbb{P}\big(a(\mathbf{z}_i) = y\big)\Bigg] \geq \mathbb{E}_{(\mathbf{x},y) \sim D(\boldsymbol{\theta}^*)}\Bigg[\frac{1}{|Z^{(t)} \setminus Z'^{(t)}|} \sum_{\mathbf{z}_i \in Z^{(t)} \setminus Z'^{(t)}} \mathbb{P}\big(a(\mathbf{z}_i) = y\big)\Bigg]$$

That is, in expectation, the responses which are selected for quality pruning are at least as correct as those which are removed by quality pruning. Therefore, in expectation across all possible generations

$\mathbf{z}_i^{(t+1)}$ with $a(\mathbf{z}_i^{(t)}) = y$,

$$\mathbb{E}_{\mathbf{z}_i^{(t+1)}}\left[\sum_{\boldsymbol{\theta}\in\Theta}\mathbb{P}\big(\mathbf{z}_i^{(t+1)}|\,\boldsymbol{\theta},\phi_i\big)\mathbb{P}(\mathbf{x}|\,\boldsymbol{\theta},\phi_i)\mathbb{P}(\boldsymbol{\theta}|\,\phi_i)\prod_{\mathbf{z}_j'\in Z'^{(t)}}^{n}\mathbb{P}\big(\mathbf{z}_j'^{(t)}|\,\boldsymbol{\theta},\phi_i\big)\right]$$

$$\geq \mathbb{E}_{\mathbf{z}_i^{(t+1)}}\left[\sum_{\boldsymbol{\theta}\in\Theta}\mathbb{P}\big(\mathbf{z}_i^{(t+1)}|\,\theta,\phi_i\big)\mathbb{P}(\mathbf{x}|\,\theta,\phi_i)\mathbb{P}(\boldsymbol{\theta}|\,\phi_i)\prod_{\mathbf{z}_j\in Z^{(t)}}^{n}\mathbb{P}\big(\mathbf{z}_j^{(t)}|\,\boldsymbol{\theta},\phi_i\big)\right]$$

$$\implies \mathbb{E}_{\mathbf{z}_i^{(t+1)}}\left[\mathbb{P}\big(\mathbf{z}_i^{(t+1)}|\,\mathbf{x},Z^{(t)},\phi_i\big)\right] \geq \mathbb{E}_{\mathbf{z}_i^{(t+1)}}\left[\mathbb{P}\big(\mathbf{z}_i^{(t+1)}|\,\mathbf{x},Z'^{(t)},\phi_i\big)\right]$$

Therefore, the probability of model $i$ generating a response $\mathbf{z}_i^{(t)}$ at time $t$, which has $a(\mathbf{z}_i^{(t)}) = y$ is greater when conditioning only on the responses selected by quality pruning, i.e., $Z'^{(t)}$. □

*Proof of Theorem 6.4.* Let $\mathbf{z}$ be a given response and $\mathbf{z}'$ be a corrected version of that response after misconception refutation, then

$$\text{KL}\big(D(\boldsymbol{\theta}|\,\mathbf{x},y),D(\boldsymbol{\theta}|\,\mathbf{z}')\big) \leq \text{KL}\big(D(\boldsymbol{\theta}|\,\mathbf{x},y),D(\boldsymbol{\theta}|\,\mathbf{z})\big)$$

That is, the distribution over concepts given the corrected response $\mathbf{z}'$ is more similar to the true distribution over concepts given the task $\mathbf{x}$ and answer $y$ compared to the original answer $\mathbf{z}$.

Similar to the case of quality pruning, we can then express the KL divergence of all responses before refutation $Z^{(t)}$, and after refutation $Z'^{(t)}$, as

$$\sum_{\mathbf{z}_i'\in Z'^{(t)}}\text{KL}\big(D(\boldsymbol{\theta}|\,\mathbf{x},y),D(\boldsymbol{\theta}|\,\mathbf{z}_i')\big) \leq \sum_{\mathbf{z}_i\in Z^{(t)}}\text{KL}\big(D(\boldsymbol{\theta}|\,\mathbf{x},y),D(\boldsymbol{\theta}|\,\mathbf{z}_i)\big)$$

$$\implies \sum_{i=1}^{n}\text{KL}\big(D(\boldsymbol{\theta}|\,\mathbf{x},y),D(\boldsymbol{\theta}|\,\mathbf{z}_i')\big) - \text{KL}\big(D(\boldsymbol{\theta}|\,\mathbf{x},y),D(\boldsymbol{\theta}|\,\mathbf{z}_i)\big) \leq 0$$

Thus, in aggregate, misconception refutation results in all responses in $Y^{(t)}$ inducing a distribution over concepts which is more similar to the distribution over concepts given the task $\mathbf{x}$ and answer $y$. As shown in the case of quality pruning, this relationship implies that at the next step of generation, model $i$ is more likely to generate $\mathbf{z}_i^{(t+1)}$ with $a(\mathbf{z}_i^{(t+1)}) = y$ when conditioning on $Z'^{(t)}$ compared with $Z^{(t)}$. □

## B  Shared Misconceptions

Latent concepts, and by extension, shared misconceptions, are quite general and may not always be human-interpretable. To better elucidate what is meant by a shared misconception, we provide an example of one possible type of misconception. In Figure 4 we see model responses to a question regarding the song "Take Me Home, Country Roads", which is a well-known song about the state West Virginia[3]. While the models identify the connection between the song and West Virginia, they each erroneously equate West Virginia and Virginia, ultimately leading to each model providing the wrong answer. In situations such as this, debate will converge to an incorrect answer due to each model sharing the same false belief.

Misconceptions can also be viewed through the lens of hallucinations. As a byproduct of erroneous training, the models in the above example have learned a false connection between two topics (Virginia and West Virginia).

## C  Experiments

**Interventions**    When combining our three interventions together, we find that first applying quality pruning, then diversity pruning, then misconception refutation, results in the best performance. In practice, we consider all previous responses up to the current round $t$ when applying pruning, i.e.,

---

[3]Models are given a question and a passage (the passage is omitted from Figure 4)

Figure 4: Example of a common misconception between models and a refutation of that misconception.

> **Question:** Is the song "Take Me Home Country Roads" about Virginia?  **Answer:** No
>
> **GPT-3.5:** Yes. The song "Take Me Home, Country Roads" is an iconic symbol of West Virginia. It describes West Virginia as "almost Heaven" and in March 2014, the song became one of the official state anthems of West Virginia. Therefore, the answer to the question is Yes, "Take Me Home, Country Roads" is about Virginia.
>
> **Llama-2:** Yes, the song "Take Me Home, Country Roads" is about Virginia. The song is associated with West Virginia, and was played at the funeral memorial of U.S. Senator Robert Byrd in July 2010. This suggests that the song is closely tied to the state of West Virginia. Therefore, the answer to the question is Yes, "Take Me Home, Country Roads" is about Virginia.
>
> **Refutation (Mistral):** West Virginia and Virginia are not the same place. West Virginia is a state located in the Appalachian region of the Eastern United States. It became a state on June 20, 1863, and was carved out of the eastern part of the former Virginia. While they share some historical and geographical similarities, they are distinct political entities.

$Z^{(0)} \cup \ldots \cup Z^{(t)}$ results in better performance. When considering all past responses during pruning, there is a potentential misconception refutation, then al that the same set of responses is picked at each round. We simply prevent the methods from selecting the same response during two consecutive rounds to avoid this issue.

**Application of Interventions in Practice**  For approximate the KL-divergence between distributions of concepts needed for our running interventions, i.e.

$$\mathrm{KL}\big(D(\boldsymbol{\theta}|\mathbf{x}), D(\boldsymbol{\theta}|\mathbf{z}_i)\big) \quad \text{or} \quad \mathrm{KL}\big(D(\boldsymbol{\theta}|\mathbf{z}_i), D(\boldsymbol{\theta}|\mathbf{z}_j)\big)$$

we use the distance between sentence embeddings of each string, $\mathbf{x}$, $\mathbf{z}_i$, and $\mathbf{z}_j$, i.e., given sentence embedding model $g$, we to approximate the above in practice via,

$$\mathrm{KL}\|g(\mathbf{x}) - g(\mathbf{z}_i)\| \quad \text{or} \quad \mathrm{KL}\|g(\mathbf{z}_i) - g(\mathbf{z}_j)\|$$

---

**Algorithm 1** Application of Combined Interventions

1: **Input:** task $\mathbf{x}$,
2: $Z_{\text{all}} = \{\}$ // Set of all responses to consider when applying interventions
3: **for** $j = 1 \ldots n$ **do**
4:    get response $\mathbf{z}_j^{(0)}$ by prompting $\text{LLM}_j$, i.e. sample $\mathbf{z}_j^{(0)}$ according to $\mathbb{P}\big(\mathbf{z}|\,\mathbf{x}, \phi_j\big)$
5:    $Z_{\text{all}}.\text{add}\big(\mathbf{z}_j^{(0)}\big)$ // Get an initial set of responses from each LLM
6: **end for**
7: **for** $t = 1 \ldots T$ **do**
8:    $Z'^{(t)} = \text{QualityPrune}\big(Z_{\text{all}}, k = 1/2|Z_{\text{all}}|\big)$ // Prune half the current responses
9:    $Z'^{(t)} = \text{DiversityPrune}\big(Z'^{(t)}, k = n\big)$
10:    $Z'^{(t)} = \text{MisconceptionRefutation}\big(Z'^{(t)}\big)$ // Set of responses to use agents to consider
11:    **for** $j = 1 \ldots n$ **do**
12:       get $\mathbf{z}_j^{(t)}$ by sampling according to $\mathbb{P}\big(\mathbf{z}|\,\mathbf{x}, Z'^{(t)}, \phi_j\big)$, for each $j$ // Updated responses
13:       $Z_{\text{all}}.\text{add}\big(\mathbf{z}_j^{(t)}\big)$
14:    **end for**
15: **end for**
16: **Return** $Z_{\text{all}}[-n:]$ // Each model's response on the last round of debate

---

**Models**  For our experiments we make use of four models: GPT-3.5, Llama-2, Llama-3, and Mistral.

For certain tasks such as BoolQ or MMLU, such as only providing "Yes" or "No" to BoolQ questions. Justifications for answers are important for both our method and regular debate, as such we set the minimum token for Flan-T5 to be 10, and we set the repetition penalty to be 1.5.

**Measuring Accuracy**  In the BoolQ, MMLU, and Math datasets, we extract the model's answers through regular expression checking and compare these extracted answers to the true answer; models are prompted to provide their final answer in the form *"Final Answer: X"*. In the TrutfhulQA dataset

Table 2: List of specific types of models used in experiments

| Model Name | Model Version | Library |
|---|---|---|
| GPT-3.5 | GPT-3.5 Turbo | openai |
| Llama-2 | Llama-2 7B Chat | huggingface |
| Llama-3 | Llama-3 8B Instruct | huggingface |
| Mistral | Mistral 7B Instruct v02 | huggingface |

model answers are taken to be their entire response, which is then judged as being correct or incorrect by a GPT-4 judge. This judge is prompted to provide a yes-no answer to the question *"Dose the answer {_answer_} accurately answer the question {_question_}?"*. We allow for models to provide answers of abstention, e.g., responding *"I do not know"*. Abstentions correspond to an accuracy of .5 in BoolQ, .25 in MMLU, 0 in Math, and are directly scored by the LLM judge in TruthfulQA.

**Target Answers**    In Figure 1 we measure the likelihood that a given model will echo a *target answer* as a function of how many other model select that target answer. Target answers are Yes in BoolQ, option A in MMLU, $correctAnswer - 30$ in Math, and a false answer in TruthfulQA. In our experiment, we elicit 20 answers from each model (GPT-3.5, Llama-2, Flan-T5) and then downsample these answers to ensure that each model receives a specific number of target answers.

**Ablation of Interventions**    Here we provide an ablation of each of our three interventions: Misconception Refutation, Diversity Pruning, and Quality Pruning. Results are shown in Table 3. From this table, we see two key takeaways. First, a combination of all three interventions achieves the highest performance in almost all cases. Second, applying intervention individually can result in worse performance (even when compared with a single model). This is expected as our interventions are designed to work together, rather than separately. Recall that when combining the interventions we first do Quality Pruning, then Diversity Pruning, and then Misconception Refutation. This ordering of interventions ensures that we first select sufficiently relevant responses (Quality Pruning), among those relevant responses we then ensure that the distribution of opinions within these responses is well balanced (Diversity Pruning), and then we lastly ensure that none of the responses contain errors or misconceptions (Misconception Refutation).

Table 3: Average accuracy for each intervention: Misconception Refutation (MR). Diversity Pruning (DP), Quality Pruning (QP), a combination of all three (Ours), and vanilla debate (Debate), for 10 rounds and 6 models. Note that the "Single", "Debate", and "Ours" columns correspond to the same columns in Table 3

| | Single | SoM | Ours | MR | DP | QP |
|---|---|---|---|---|---|---|
| | **BoolQ** | | | | | |
| 6× GPT-3.5 | $.80_{\pm.014}$ | $.84_{\pm.012}$ | $\mathbf{.85}_{\pm.012}$ | $.83_{\pm.013}$ | $.84_{\pm.012}$ | $.84_{\pm.012}$ |
| 6× Llama-3 | $.76_{\pm.014}$ | $.78_{\pm.013}$ | $.78_{\pm.013}$ | $.78_{\pm.013}$ | $.76_{\pm.013}$ | $.77_{\pm.013}$ |
| 6× Llama-2 | $.67_{\pm.017}$ | $.68_{\pm.017}$ | $.70_{\pm.016}$ | $.67_{\pm.017}$ | $.69_{\pm.016}$ | $.73_{\pm.016}$ |
| 6× Mistral | $.80_{\pm.014}$ | $.82_{\pm.013}$ | $\mathbf{.85}_{\pm.012}$ | $.83_{\pm.013}$ | $.81_{\pm.014}$ | $.82_{\pm.013}$ |
| | **MMLU** | | | | | |
| 6× GPT-3.5 | $.73_{\pm.014}$ | $.74_{\pm.015}$ | $\mathbf{.79}_{\pm.014}$ | $.75_{\pm.015}$ | $.72_{\pm.016}$ | $.73_{\pm.016}$ |
| 6× Llama-3 | $.67_{\pm.016}$ | $.70_{\pm.015}$ | $.75_{\pm.014}$ | $.71_{\pm.015}$ | $.68_{\pm.016}$ | $.68_{\pm.016}$ |
| 6× Llama-2 | $.41_{\pm.017}$ | $.47_{\pm.018}$ | $.52_{\pm0.18}$ | $.50_{\pm0.18}$ | $.40_{\pm0.17}$ | $.43_{\pm0.17}$ |
| 6× Mistral | $.66_{\pm.016}$ | $.65_{\pm.016}$ | $.66_{\pm.016}$ | $.66_{\pm.016}$ | $.58_{\pm.018}$ | $.62_{\pm.017}$ |
| | **TruthfulQA** | | | | | |
| 6× GPT-3.5 | $.61_{\pm.033}$ | $.63_{\pm.032}$ | $\mathbf{.69}_{\pm.030}$ | $.65_{\pm.032}$ | $.58_{\pm.034}$ | $.62_{\pm.034}$ |
| 6× Llama-2 | $.47_{\pm.034}$ | $.52_{\pm0.35}$ | $.55_{\pm.034}$ | $.53_{\pm.034}$ | $.46_{\pm.034}$ | $.44_{\pm.034}$ |
| 6× Llama-3 | $.53_{\pm.035}$ | $.55_{\pm.032}$ | $.55_{\pm.032}$ | $.55_{\pm.032}$ | $.53_{\pm.032}$ | $.54_{\pm.032}$ |
| 6× Mistral | $.48_{\pm.034}$ | $.51_{\pm.035}$ | $.53_{\pm.034}$ | $.53_{\pm.034}$ | $.49_{\pm.034}$ | $.47_{\pm.034}$ |
| | **MathQ** | | | | | |
| 6× GPT-3.5 | $.53_{\pm.035}$ | $.88_{\pm.016}$ | $\mathbf{.93}_{\pm.01}$ | $.92_{\pm.01}$ | $.85_{\pm.016}$ | $.86_{\pm.016}$ |
| 6× Llama-2 | $.11_{\pm.013}$ | $.13_{\pm.014}$ | $.19_{\pm.015}$ | $.18_{\pm.015}$ | $.13_{\pm.014}$ | $.13_{\pm.014}$ |
| 6× Llama-3 | $.25_{\pm.016}$ | $.33_{\pm.017}$ | $.48_{\pm.018}$ | $.49_{\pm.018}$ | $.32_{\pm.017}$ | $.32_{\pm.017}$ |
| 6× Mistral | $.13_{\pm.013}$ | $.19_{\pm.014}$ | $.18_{\pm.014}$ | $.17_{\pm.014}$ | $.15_{\pm.014}$ | $.16_{\pm.014}$ |

**Resource Details** For all experiments, we use one Nvdida Tesla V100 GPU and one Intel 32-core CPU. For inference with non-API models (i.e., Llama-2, Llama-3, and Mistral, we use the VLLM library Kwon et al. [2023]). Ten rounds of debate with six models and 3,000 questions has a mean completion time of 4 hours for Llama-2, Llama-3, and Mistral. For GPT-3.5, we use the open library, ten rounds of debate with six models and 3,000 questions, has a mean completion time of 12 hours.

## C.1 Prompt Examples

We provide example prompt templates for the BoolQ dataset. Prompt templates for other datasets are similar but have changes to reflect the different task types (e.g., multiple-choice answers in MMLU compared to yes-no answers in BoolQ).

**Round 0, or No Debate:**

```
prompt = You will be given a yes-no question which is based on a passage.
         You should use the passage to help you answer the question.
         You should give a brief justification for your answer,
         and you must provide a final answer of either Yes or No.
         \n Question: {_QUESTION_}
         \n Passage: {_PASSAGE_}
```

**Debate with Round** $> 0$

```
prompt = Several other models have provided responses to a yes-no question,
         below are their responses:
         \n Model 1: {_RESPONSE[1]_}
         .
         .
         .
         \n Model n: {_RESPONSE[n]_}
         \n You should consider these responses when answering the following yes-no
         question which is based on a passage.
         You should use the given responses and the passage to help you answer the question.
         You should give a brief justification for your answer, and you must provide a
         final answer of either Yes or No.
         \n Question: {_QUESTION_}
         \n Passage: {_PASSAGE_}
```

**Misconception Refutation (Identifying Misconceptions)**

```
prompt = I would like you to evaluate an answer to a question based on a passage.
         Please evaluate this answer and identify any errors, misconceptions,
         or inconsistencies with the passage.
         If you identify any such errors, please provide
         a short list of specific details and briefly discuss how the misconceptions
         can be fixed.
         \n Question: {_QUESTION_}
         \n Passage: {_PASSAGE_}
         \n Answer to Evaluate Answer: {_GIVEN_ANSWER_}
```

**Misconception Refutation (Fixing Misconceptions)**

```
prompt = I would like you to make corrections to a response.
         You will be given a yes-no question based on a passage, a response to
         that question, and a list of possible issues with the response.
         I want you to provide a corrected version of the response based on the
         list of possible issues.
         You should make as few changes as possible.
         \n Question: {_QUESTION_}
```

```
\n Passage: {_PASSAGE_}
\n Response to Correct: {_RESPONSE_}
\n Possible Issues: {_LIST_OF_ISSUES_}
```

**Targeted Answer (Advocating for a Specific Answer)**

```
prompt = You will be given a yes-no question which is based on a passage.
         You should use the passage to provide an answer of {_TARGET_ANSWER_}.
         You should give a brief justification for that answer,
         and you must provide a final answer {_TARGET_ANSWER_}.
         \n Question: {_QUESTION_}
         \n Passage: {_PASSAGE_}
```

