# OpenReview forum: "Multi-LLM Debate: Framework, Principals, and Interventions"
_NeurIPS.cc/2024/Conference — NeurIPS 2024 poster_

### Official Review · Reviewer_1QQG · 2024-06-24

**Soundness:** 2
**Presentation:** 3
**Contribution:** 2
**Rating:** 4
**Confidence:** 5

**Summary:**

The paper tackles the issue of echo chambers in multi-agent debate (MAD), a critical area that has not been sufficiently explored theoretically in current MAD research as the authors claimed. The authors propose three interventions – diversity pruning, quality pruning, and refuting misconceptions – to address this problem.

**Strengths:**

Positive aspects:

1. The paper addresses the important issue of echo-chamber effects in multi-agent debates.
2. The authors propose theorems to clarify concepts, attempting to provide a theoretical foundation for their work.
3. The paper is written in an accessible manner, making it easy to understand for readers.

**Weaknesses:**

Major critiques in several areas:

A. Theoretical Issues:

A.1 Underdeveloped theoretical foundation: While the authors propose theorems, many appear trivial or state obvious conclusions (e.g., Theorems 5.1 and 5.2).

A.2 Ambiguous parameter representation: The representation of θ in Assumption 4.1 is unclear, raising questions about how model parameters are represented and changed.

A.3 Shared latent space issues: Lemma 4.2 doesn't adequately address how different LLMs would share their latent space.

B. Methodological Concerns:

B.1 Simplistic multi-agent dialogue setup: The model fails to fully leverage the potential of a multi-LLM ensemble, using it more like traditional ensemble schemes.

B.2 Unclear latent concept model: The origin and nature of latent concepts in Section 2 lack clarity.
Lack of quantifiability: The formulation in Section 4 cannot be quantified, leading to reliance on heuristics.

C. Implementation and Experimental Issues:

C.1 Disconnect between theory and implementation: Section 6 presents questionable formulations, particularly in the use of KL divergence and reliance on sentence embedding as a proxy.

C.2 Outdated model selection: Experiments use older models instead of more recent, capable ones.
Limited experimental insights: Results only show significant improvement in the first round of voting.

C.3 Questionable example: The example in Section 5.2 doesn't effectively illustrate the point about misconceptions.

Conclusion: While the paper addresses an important issue and attempts to provide theoretical grounding, it falls short in several areas. The multi-LLM debate framework presented doesn't demonstrate its full potential for sophisticated decision-making tasks. The intervention strategies and control parameters lack necessary granularity. Despite the paper's accessibility, the gap between theoretical concepts and practical application remains significant, with many proofs being trivial or of limited practical value.

Recommendation for future work: Focus on developing more sophisticated decision-making tasks, refine intervention strategies, and use state-of-the-art language models for experiments. Additionally, strengthen the connection between theoretical concepts and practical implementations to enhance the paper's overall impact and relevance.

**Questions:**

1. Suppose KL represents KL divergence. KL divergence is asymmetric and unbounded. Why not use other metrics?

2. In the experiment section, would you consider adding ablation tests to determine which of your intervention strategies are effective?

3. Can you clarify how you pass the distribution of latent concepts between LLMs?

4. Please find more related papers in multi-LLM debate or dialogue.  You missed a couple of key papers.

**Limitations:**

NO! Limitation section missing.

The authors on the checklist state that they submitted limitations, but they did not.
This paper could have been desk-rejected.

---

> ### Author Rebuttal · Authors · 2024-08-06
>
> ### A Theoretical Issues:
>
> **[Theoretical Results]** We aim to provide theoretical results that help explain why specific behavior would be observed in the debate process. For example, Theorem 5.2 is intended to help understand why one would observe tyranny of the majority LLM debate. While not surprising to observe tyranny of the majority in debate, we are first to establish this observation rigorusly. Our result complements the main findings of [1], which outlines a similar phenomenon in in-context-learning; when many of the examples share an underlying pattern, models are more likely to give responses that also have that shared underlying pattern (Figure 1 in [1] outlines the basic idea). We don't wish to overrepresent the impact of our results, but we do genuinely believe that the theorems we provide are informative.
>
> **[Parameter Representation and $\theta$]** We agree that in isolation, the representation of $\theta$ in Assumption 4.1 is unclear. We discuss the role of $\theta\in \Theta$ in detail, starting on line 89. To address this, we propose to change the language of Assumption 4.1 to read:
>
> **Assumption 4.1** For a given latent concept sapce $\Theta$, the probability of generating response $\mathbf{z}_i^{(t+1)}$ is conditionally independent of both the responses $Z^{(t)}$ and the task $x$ when given any concept $\theta\in\Theta$ and model parameters $\phi_i$, i.e., $\mathbb{P}\big(\mathbf{z}_i^{(t+1)}|\theta, \mathbf{x}, Z^{(t)}, \phi_i\big) = \mathbb{P}\big(\mathbf{z}_i^{(t+1)}|\theta, \phi_i\big)$.
>
>
>
> **[Communication]** Models communicate with each other only through tokens and do not share their latent space.
>
> ### B Methodological Concerns
>
> **[Simplistic Multi-Agent Dialogue]** We agree that debate as a paradigm fails to fully leverage the potential of a multiple LLMs. Our main goal with this paper is to provide a more rigorous formalization of debate. Given the growing number of papers focusing on multi-LLM debate (which all use a very similar setup to ours), we believe that our work is valuable.
>
> We would also like to point out that debate is quite different from traditional ensemble methods. Stoping debate at round $t=0$ would be equivalent to a traditional ensemble. With that said, we would be very interested in seeing general multi-LLM interactions explored further, this is outside the scope of our work.
>
>
> **[Latent Concepts and a Reliance on Heuristics]** We follow the latent concept model of [1, 2]. The true latent concept space will not be accessible; we use proxies such as sentence embeddings. We fully agree that our method's weakness is its reliance on heuristics. However, our theory informs the development of these heuristics. In our experiments, these heuristics are effective.
>
> ### Implementation and Experimental Issues
>
> **[Theorey and Implementation]** We use our theory to help guide and motivate the development of our heuristics. While we use proxies for statistical distance (namely, the distance between sentence embeddings), the choice of this proxy, as well as what we do with that proxy, are the direct result of our theoretical results. Beyond just informing the heuristics, our theoretical results also help provide some intuition as to why these heuristics work. In particular, it may not be obvious a priori why diversity pruning effectively reduces tyranny of the majority (as demonstrated in Figure 2). Both theorem 5.2 and the formulation diversity pruning (given on line 232) help explain this effectiveness more intuitively than just empirical observation alone.
>
>
> **[Model Choice]** We use Llama-3, Llama-2, Mistral-v0.2, and GPT-3.5. Both Llama-3 and Mistral-v0.2 are state-of-the-art models. While we agree that Llama-2 (which has been replaced by Llama-3) and GPT-3.5 are not state-of-the-art, they are still quite capable models. Our experiments show that our method is effective on a diverse set of models, including quite capable models such as Llama-3.
>
> **[Improviemnt in the First Round]** Our method achieves improvement over the baseline across all rounds (not just the first round). For both vanilla-debate and our method, the largest average gain is at round $t=1$. Both methods continue to improve over subsequent rounds  (Figure 4). Ultimately, we observe significant improvement over vanilla-debate (in most cases) even after 10 rounds of debate (Table 1).
>
>
> **[Example about misconceptions]** Our example of misconceptions indicates that misconceptions represent a particular type of error, specifically erroneous associations between more fundamental underlying ideas (concepts).
>
> **[Choice of KL Divergence]** While we could choose other distance metrics, we choose KL divergence specifically to capture the surprise (in a technical sense of the word) when using one of the responses $\mathbf{z}_i$ as a predictor of $\theta$ compared to using the original task $\mathbf{x}$ as a predictor of $\theta$. Under this interpretation, the asymmetry of KL is desirable for quality pruning and misconception refutation. In the case of diversity pruning, because we sum over all pairs of $\mathbf{z}_i, \mathbf{z}_j$, a symmetry of KL is lost. We agree that the unbounded nature of KL can be undesirable.
>
> **[Ablation of Each Intervention]** We provide an ablation of each intervention in Table 3 of the supplement and point to this table on line 314 of the main text.
>
> **[Related Work]** There are quite a few relevant papers that have come out after the submission deadline that we plan to add. If you know of any other papers that we have missed, please let us know. We find this area deeply intriguing and would be very grateful for pointers towards any interesting work that we may have overlooked.
>
>
> **References**
>
> [1] Xie, Sang Michael, et al. "An explanation of in-context learning as implicit Bayesian inference." ICLR (2022).
>
> [2] Hui Jiang. A latent space theory for emergent abilities in large language models. arXiv preprint 354 arXiv:2304.09960, 2023.

---

> > ### Comment · Reviewer_1QQG · 2024-08-08
> > **related work, etc.**
> >
> > Thank you for providing clarifications. Because of the tight schedule, let me provide a textbook relevant to your work for consideration and discussion. The book was used in a university class and was known in the NLP community. Its Chapters 4 to 9, and especially chapter 5 and 6 are highly relevant to this paper.  E.g., chapter 6 justifies using cross-entropy, earth-mover, mutual information and other metrics in addition to the KL.
> >
> > https://www.amazon.com/dp/B0D8NGQZFR

---

> > > ### Author Response · Authors · 2024-08-09
> > > **Authors' Response**
> > >
> > > Thank you for the pointer to this textbook! We were not aware of this book at the time of submission. We agree that this text is highly relevant to any paper investigating multi-LLM debate. We plan to cite this book and make special mention of chapters 5 and 6; specifically, we propose to add the following text to our related work section:
> > >
> > > > “An overview of debate techniques and analysis can be found in [Chang 24]. Chapter 5 outlines a debate procedure that dynamically controls the amount of disagreement between debating LLMs. Chapter 6 outlines the way in which LLM responses change during the debate process; distributions over responses tend to converge (approximately) to consensus as debate rounds increase.”
> > >
> > > As you point out, chapter 6 offers justification for studying distributions of LLM responses via several metrics including KL. For clarity we would like to mention that we study distributions over latent concepts (see Lines 233, 237 and 260 of our paper) rather than distributions over LLM responses (as is the case in [Chang 24]). Further differentiating our work is the use of these distributions to then optimize the debate procedure.
> > >
> > > If we can provide any additional clarification, please let us know.
> > >
> > > [1] Edward Y. Chang (2024). The Path to Artificial General Intelligence –- Insights from Adversarial LLM Dialogue.

---

> ### Comment · Reviewer_1QQG · 2024-08-10
>
> Dear Authors: Thank you for your response.  I will adjust the rating accordingly.
>
> Dear Area Chair,
>
> Please determine whether the omission of the "limitations" section in the submitted checklist is a significant issue. Consistency in review policy is crucial for fairness, especially given that both of our NeurIPS submissions were desk-rejected due to checklist-related issues.
>
> Thank you.
>
> -------------------
>
> Dear Authors,
>
> According to the NeurIPS policies, any violence of formatting instructions (as indicated in the Call for Papers: https://neurips.cc/Conferences/2024/CallForPapers -> Formatting instructions) results in a desk rejection. Your paper was rejected since the NeurIPS paper checklist is part of these formatting instructions (see  https://neurips.cc/Conferences/2024/CallForPapers -> Formatting instructions -> Point 3). Unfortunately, we are unable to allow further submissions or amendments to submissions as the system is now closed.
>
> We are aware that this is not a decision you expected, but we must follow the NeurIPS policies. The decision is final and cannot be reverted.

---

> > ### Comment · Area_Chair_1rCB · 2024-08-13
> >
> > Dear reviewer,
> > Thank you for carefully reading the paper and raising the issue. However, it is not your role to decide whether the paper gets desk-rejected or not. I'll check with SAC about this, but I do not think this is a serious enough issue for desk rejection.

---

> ### Author Response · Authors · 2024-08-12
> **Author Followup**
>
> We are sorry to hear the unfortunate desk rejections from your end, and thank you for reminding us of this. In the future, we will pay closer attention to it.
>
> At the time of submission, we had a different interpretation of the limitation requirement. While a separate section is encouraged, we thought the paper flows better when we discuss the limitations right in the main text, and labeling Yes simply states that we are aware of our limitations and have discussed them in the main text. (in the checklist, we added the justification that "Justification: We mention the limitation of our approach in the conclusion section and experiment section").
>
> That said, we would like to be clear that we fully agree with the reviewer's comments regarding our discussion of limitations; our initial discussion of limitations was insufficient and should be provided in its own section (please see our general response for our proposal).

---

> ### Comment · Reviewer_1QQG · 2024-08-13
> **Misunderstanding**
>
> I pasted the statement of rejecting my papers.  Of course I have no authority what-so-ever to make that decision.
>
> Good luck!

---

### Official Review · Reviewer_C8as · 2024-07-11

**Soundness:** 4
**Presentation:** 3
**Contribution:** 3
**Rating:** 8
**Confidence:** 4

**Summary:**

The main questions, that is tackled in this paper is: How can the debate of LLMs be influenced to ensure the best possible outcome. This question is broken down into three parts.

The first part is a theoretical model of debate between LLMs. The model is very intreaging. From the presentation it is not clear, why it should only be limited to the debate of LLMs. It would be very interesting to see, how the presented model of debate relates to models of debate from other fields, e.g. sociology.

The second part describes three methods of interfering with the outputs of the models to ensure that they do not converge to a suboptimal solution.

The third part shows the results of experiments.

**Strengths:**

The models, assumptions, theorems and conclusions are very well and are also easy to follow for readers who may not be so familiar with the mathematics.

I would like to see, if the model of debate could also be applied to other settings.

The experiments confirm the theory, i.e. the methods of interfering show a clear improvement in the performance of the systems.

**Weaknesses:**

There are some typos in the document, that need to be cleaned up.

The introduction to section 5 seems incomplete. Line 166 ends within a sentence. There seems to be a grammar mistake and a ) too much in line 158.

**Questions:**

I had a hard time understanding, why Diversity Pruning is referred to as maximizing the mutual information. Intuitively, maximizing mutual information would mean, that by knowing X you know everything or a lot about Y. Here, you are doing to the opposite. You are maximizing the difference of D(Theta|z_i) and D(Theta|z_j), hence, knowing z_i should not give you much information on how z_j influences Theta. I would appreciate, if you could help me understand my misunderstanding.

**Limitations:**

The authors claim that they have addressed the limitations. I was not able to find that section. Please point out, where you discuss the limitation of your models and interventions.

---

> ### Author Rebuttal · Authors · 2024-08-06
>
> **[Applying Debate to Other Settings]** We think that applying debate to settings beyond QA tasks would be very interesting. As of now the majority the works on debate focus on QA tasks (while some investigate translation). We hope that our framework inspires further generalization to other settings.
>
> **[Typos and Grammar]** Thank you for pointing these out. We will address these in the final version of our paper.
>
>
> **[Diversity Pruning and Mutual Information]** It would be more accurate to say that we are maximizing the amount of unique information present in the responses $Z'^{(t)}$, i.e., maximizing the information entropy present in $Z'^{(t)}$. Our use of the term "mutual information" arose during editing. Initially, we had said, "the amount of unique information mutually present" and then edited this phrase down to “mutual information” on lines 35 and 232, without realizing that we had inadvertently invoked a technical term conveying something other than our intention. This is an oversight on our part; we will modify the language around diversity pruning from "mutual information" to "information entropy".

---

> > ### Comment · Reviewer_C8as · 2024-08-12
> > **Answer**
> >
> > Thank you very much for the rebuttal and addressing my comments. I am looking forward to the final version of the paper.

---

### Official Review · Reviewer_6kQa · 2024-07-12

**Soundness:** 3
**Presentation:** 2
**Contribution:** 3
**Rating:** 6
**Confidence:** 4

**Summary:**

The paper presents a framework for multi-LLM debate, gives a detailed overview on drawbacks within multi-LLM debate and proposes several interventions to improve debates. The authors show the effectiveness of their interventions in several experiments on four benchmarks.

**Strengths:**

The task of multi-LLM debate is interesting, and important to further improve LLMs responses. The authors gave a detailed overview on effects that can occur within multi-LLM debate and proposed effective interventions to improve debate  on four benchmarks.

**Weaknesses:**

The related work section would be more readable, if the author's names where mentioned instead of the reference number. Like in: "[19] focuses on debate when each model..."
KL denotes the Kullback–Leibler divergence, which should be noted, even if it seems obvious.
There are several Mistral models. I would specify the model further (Mistral 7B).
Figure 3 is a bit too small and the two lines are the same color, so it's hard to tell them apart.
You chose Debate as the main baseline tool. A detailed explanation of the method to understand the differences of the approaches is missing.
- The method and task are both called "debate.", which is confusing.

small errors:
- line 82 should be "that agent i provides response zi(t+1), is given by"..
- no error, just a small suggestion: in line 86 write (1) and (2) instead of 1) and 2) which makes it more readable in my opinion
- line 94: task x, and its associated answer y (singular) else I would not understand why x denotes multiple tasks?
- line 189: do you mean with the property instead of propriety?
- it seems that Diversity Pruning should maximize the KL-divergence and not the mutual information as stated in lines 232 and 233. In the diversity pruning we are maximizing over the answers Z so it should be Z ⊂ Z(t) instead of Y.
- What is S (z ∈ S) in the formula for misconception refutation?
- line 292: Z(0) instead of Z0)
- Figure 3: small error underneath the graphic: you wrote pairwise similarity twice

**Questions:**

- Maybe I overlooked the explanation in the paper, but what does alpha denote in Formula (3)?
- How does the misconception refutation work exactly? How should a model be able to identify misconceptions in the answer of another model?

**Limitations:**

An explicit sections that mentions limitations of the work is missing.

---

> ### Author Rebuttal · Authors · 2024-08-06
>
> **[Citations and Specificity]** Thank you for pointing this out. We agree with both points regarding the related work. We will add named citations rather than numbered citations. The specific model versions are provided in Table 2 of the supplement, but we will add this information to line 280 of the main body.
>
> **[Figure 3]** We will make this figure large and change the dashed green line to a dashed blue line to further distinguish these values.
>
> **[Difference Between Debate and Our Method]** Our method (namely our three interventions) is an in-processing modification to the original debate procedure in which we leverage a latent concept space $\Theta$ to modify the responses $Z^{(t)}$ before these responses are used at the next round $t+1$. We provide an overview of the vanilla debate procedure (proposed in [Du, Yilun, et al.]) on line 73 of our paper. To help clear up this point, we propose to cite [Du, Yilun, et al.] at the beginning of our discussion on debate (line 73), and to state after line 88 that “The key distinction between our method and vanilla debate is that we will leverage a latent concept space (discussed next) to modify the responses $Z^{(t)}$ in-between round)”.
>
>
>
> **[Method and Task are Both Called Debate]**  We agree that overloading the term debate can cause confusion. We propose the following fix: we will refer to the debate baseline [Du, Yilun, et al.]  used in our experiments as "vanilla-debate" (VD), and we will refer to our method as either "Ours" or "Debate via Latent Concepts" (DLC).
>
> **[Small Errors]** Thank you for pointing these out. We will make these fixes in the final version of our manuscript.
>
> **[Role of $a$ in Formula (3)]** In Formula (3), the variable $a$ is an "answer extraction" function explained on line 80 of our paper. If $\mathbf{z}$ is a string, then $a(\mathbf{z}))$ is the extracted answer from that string. For example, suppose the question is "What color is the sky?" and $\mathbf{z} =$ "During the day, the sky is blue.", then $a(\mathbf{z}) =$ "blue." In practice, $a$ is implemented either through regular expression matching (as in the case of BoolQ, MMLU, and MathQ) or an LLM judge (as is the case in TruthfulQA).
>
>
> **[Misconception Refutation]** In practice, misconception refutation works as follows. A model $f_1$ provides a response $\mathbf{z}$. A second model $f_2$ is then asked to identify a list of misconceptions in $\mathbf{z}$. The first model $f_1$ is then reprompted for an updated response that addresses the misconceptions raised by $f_2$ ($f_1$ and $f_2$ can be the same model). The intuition for why this works is very similar to why methods such as self-refinement are effective [Madaan, Aman, et al.] (also called self-reflection); judging is often an easier task than generating. However, it is also important to note that there is a limit to how well LLMs can self-identify their own mistakes as pointed out by [Tyen, Gladys, et al.], LLMs can have a difficult time identifying their own errors in reasoning tasks, but (as the title suggests) can fix erroneous reasoning steps when being told that a given step is wrong (even without being told what the correct step should have been). This observation might indicate that misconception refutation scales in the effectiveness of $f_2$ (responsible for misconception identification) more than the effectiveness of $f_1$.
>
>
> **References**
>
> [1] Du, Yilun, et al. "Improving factuality and reasoning in language models through multiagent debate." arXiv preprint arXiv:2305.14325 (2023).
>
> [2] Madaan, Aman, et al. "Self-refine: Iterative refinement with self-feedback." Advances in Neural Information Processing Systems 36 (2024).
>
> [3] Tyen, Gladys, et al. "LLMs cannot find reasoning errors, but can correct them!." arXiv preprint arXiv:2311.08516 (2023).

---

> > ### Comment · Reviewer_6kQa · 2024-08-11
> >
> > Thank you for considering all my remarks for adapting your final manuscript.
> > I will adjust my rating accordingly. I am just confused about the comment from reviewer 1QQG who pointed out that the paper should have been desk rejected because of the missing limitations sections. Is the last comment an official response from the area chairs?

---

> > > ### Author Response · Authors · 2024-08-12
> > > **That last comment was not for this paper, but copied from somewhere else**
> > >
> > > Thank you for your feedback.
> > >
> > > The last comment from reviewer 1QQG is **not** for this paper. We believe it is a paragraph copied from the reviewer either from their end or from somewhere else.
> > >
> > > As we replied to reviewer 1QQG, at the time of submission, we had a different interpretation of the limitation requirement (Does the paper discuss the limitations of the work performed by the authors?). While a separate section is encouraged in the guideline, we thought the paper flows better when we discuss the limitations right in the main text, and labeling the answer as "Yes" simply states that we are aware of our limitations and have discussed them in the main text. Indeed, in the checklist, we had the following justification: "Justification: We mention the limitation of our approach in the conclusion section and experiment section".
> > >
> > > Please let me know if there are further questions.

---

### Official Review · Reviewer_iBgJ · 2024-07-13

**Soundness:** 2
**Presentation:** 3
**Contribution:** 2
**Rating:** 6
**Confidence:** 3

**Summary:**

In this paper, the authors establish a theoretical framework for multiagent debate. The authors find that LLMs are susceptible to issues such as leaning toward majority opinion (tyranny of the majority) and error expansion during debates (shared misconceptions between models). Thus, the authors propose three types of interventions, including debate selection and debate modification to mitigate these issues. Results show that LLMs improve on 4 common datasets, BoolQ, MMLU, TruthfullQA, Math.

**Strengths:**

1. Multiagent debate shows promising improvement of LLMs. However, most of the papers are on the empirical level. This paper provide a theoretical framework that gives a more accurate analysis of what we should focus on.
2. The phenomena of the tyranny of the majority and shared misconceptions between models are very interesting. They show how large language model bias occurs when debating without human participation.
3. The empirical results prove the theory and improve the original debating framework by a large margin.

**Weaknesses:**

1. Some assumptions seem too strong. It is assumed that each agent can only see one round of history. However, empirically, the agent can see all his own dialogues. For Asumpotion 4.1, how could the model respond without the reaction of other models? For instance, other models may react with opposite results.

2. Lacks some details on experiments. How do you force the model to affirm and then prompt m of the models (out of 11) to provide responses (Line 287)? Besides, for Figure 2, what is the dataset used, and is it averaged across all 4 datasets?

3. Since debating is ready API costly, I would expect a time estimation for the proposed algorithm on one sample and how it compares with the original debating.

**Questions:**

Some presentation typos:

line 292 $Z^{(0)}$

Figure 3 caption is repeating

**Limitations:**

I encourage the authors to write limitation and broader impacts sections independent of the main paper.

---

> ### Author Rebuttal · Authors · 2024-08-06
>
> **[Theoretical Assumptions]** While we agree that we make some stylized assumptions, we believe that these assumptions are reasonable in the context of debate and LLMs. For example, we formulate debate in a Markovian manner (i.e., each agent can only see one round of history). This Markovian property is not unique to our paper and is common in most papers on LLM debate (e.g., [Du, Yilun, et al.]). It would be interesting to consider cases where agents have access to longer histories, especially with the rapidly growing context windows of modern models (e.g., 128K token window of the newly released Llama 3.1).
>
> As for Assumption 4.1, we would like to provide some clarification. At each round $t$, the question $\mathbf{x}$ and the previous responses $Z^{(t-1)}$ are formatted into a prompt, which is passed to the model. Assumption 4.1 says that the model's generation at time $t$ (namely $\mathbf{z}^{(t)}$) is independent of the question and the other models' responses at time $t$ **given** the latent embedding $theta$ and model parameters $\phi$. This assumption can be thought of as saying that once a prompt has been passed through the encoder of the model, the original prompt (i.e., $\mathbf{x}$ and $Z^{(t)}$ in our case) is no longer needed.
>
>
> **[Majority Answers and Figure 2]**
> For the results in Figure 2, we force the models to provide specific answers using special target prompts (an example for BoolQ is provided below). We design these prompts to be as similar to the original prompts as possible (see line 576 of the supplement for the original BoolQ prompt).
> We use the targeted prompt 20 times for each target answer to obtain a diverse set of responses. Any response that does not provide the target answer is removed. We will add this explanation to the experiment section (after the first sentence of line 285) and the following example prompt to Section B.1 of the supplement (after line 617).
> ```
> You will be given a yes-no question which is based on a passage.
> You should use the passage to provide an answer of {_TARGET_ANSWER_}.
> You should give a brief justification for that answer,
> and you must provide a final answer {_TARGET_ANSWER_}.
> \n Question: {_QUESTION_}
> \n Passage: {_PASSAGE_}
> ```
> The values shown in Figure 2 are averages over all questions in each dataset (see line 273 for these values). We show one plot for each dataset, denoted by each plot's title in Figure 2.
>
> If anything else is unclear about the experimental design, please let us know; we are always eager to improve the readability of our paper.
>
>
> **[Running Time]** This is a good point; debate as an inference procedure can be costly, and our method introduces additional overhead. We believe that the performance gain from our method justifies the increased inference time. Below is a table showing the average run time for a single question (in seconds) for both the vanilla debate method and each of our interventions. We use 6 models and report the average time to complete 1 question, averaged over 3000 questions from BoolQ. The comparisons are vanilla debate (VD), diversity pruning (DP), quality pruning (QP), misconception refutation (MR), and a combination of all three interventions outlined by Algorithm 1 (All 3).
>
> | Model            | VD      |     DP   |     QP  |    MR   | All 3    |
> | ---------------- | ------- | -------- | ------- |-------- | -------- |
> | $6\times$GPT-3.5 |    2.3  |   2.6    |    2.5  |   4.8   |   4.3    |
> | $6\times$Llama-3 |    5.2  |   4.6    |    3.8  |   10.3  |   7.1    |
> | $6\times$Llama-2 |   13.8  |   11.2    |    6.6  |   27.1  |   18.0   |
> | $6\times$Mistral |    6.3  |   4.9    |    4.5  |   12.8  |   9.4    |
>
> When combining all three of our interventions, we see that the additional overhead compared to vanilla debate is not too severe. This is due primarily to the fact that diversity pruning and quality pruning remove responses from $Z^{(t)}$, which shortens the prompt lengths at round $t+1$, which in turn speeds up the inference time of each model. We will add this table to Section B of the supplement and reference/discuss this table on line 312 of our experiments section.
>
> For Llama-2, Llama-3 and Mistral, we use a single Tesla-v100-32GB GPU, an Intel i9 CPU, and the VLLM acceleration library (https://docs.vllm.ai/en/latest/). For GPT-3.5 we use the openai api and use the AsyncOpenAI class to handel the prompts in batches of 500 (https://github.com/openai/openai-python).
>
> **References**
> [1] Du, Yilun, et al. "Improving factuality and reasoning in language models through multiagent debate." arXiv preprint arXiv:2305.14325 (2023).

---

> > ### Comment · Reviewer_iBgJ · 2024-08-12
> >
> > Thanks for the point-by-point in-depth analysis and thoughtful explanations. I appreciate that my concerns on W2 and W3 are mostly addressed. I encourage the authors to include the analysis in the final version. Thanks!

---

### Author Rebuttal · Authors · 2024-08-06

# General Rebuttal

Thank you for taking the time to provide feedback on our work.

**[Limitations]** While we discuss our work's limitations within the main body, we realize that this discussion is insufficient and needs to be more overt, focused, and in its own section. We propose adding the following section right before our conclusion section. We will move Figure 1 to the supplement to make room for this additional section.

**Limitations and Impact** While we do aim to address some of the fundamental issues of multi-LLM debate, such as tyranny of the majority, there are several factors that one needs to consider when adopting our framework or using our interventions. Firstly, our theoretical results leverage a latent concept space $\Theta$, which may not be accessible in practice, necessitating the use of proxies such as sentence embeddings. This can particularly limit the applicability of quality pruning and diversity pruning; these two interventions are less effective in domains where sentence embeddings are less meaningful (e.g., arithmetic questions). Additionally, these interventions can increase the inference time of debate (this is particularly true for misconception refutation).  While our work aims to provide some insights into the debate procedure, there is much that still needs to be explored.

---

### Decision · Program_Chairs · 2024-09-25

**Decision:**

Accept (poster)

**Comment:**

The paper presents a theoretical framework for studying multi-LLM debate, and further proposes two types interventions to debate processes. The framework establishes principles of debates, including the diversity of model responses and shared misconceptions among the models. The first type of intervention is pruning, which filters responses to be included in the response, and the second type of intervention is to minimize misconception, verifying and editing previously generated answers.

Lastly, they show empirical results of using their protocol compared to prior work [Du et al 23]. The reviewers appreciate the topic studied (reviewer 6kQa, iBgJ) and the empirical gains of suggested interventions (iBgJ, C8as). The authors should include discussions from the rebuttal period, such as analysis on how their framework increases inference costs and improve clarity of the paper.